# THE EIGENLEARNING FRAMEWORK: A CONSERVATION LAW PERSPECTIVE ON KERNEL REGRESSION AND WIDE NEURAL NETWORKS

## ABSTRACT

We derive a simple unified framework giving closed-form estimates for the test risk and other generalization metrics of kernel ridge regression (KRR). Relative to prior work, our derivations are greatly simplified and our final expressions are more readily interpreted. In particular, we show that KRR can be interpreted as an explicit competition among kernel eigenmodes for a fixed supply of a quantity we term "learnability." These improvements are enabled by a sharp conservation law which limits the ability of KRR to learn any orthonormal basis of functions. Test risk and other objects of interest are expressed transparently in terms of our conserved quantity evaluated in the kernel eigenbasis. We use our improved framework to: i) provide a theoretical explanation for the "deep bootstrap" of Nakkiran et al. (2020), ii) generalize a previous result regarding the hardness of the classic parity problem, iii) fashion a theoretical tool for the study of adversarial robustness, and iv) draw a tight analogy between KRR and a well-studied system in statistical physics.

## 1 INTRODUCTION

Kernel ridge regression (KRR) is a popular, tractable learning algorithm that has seen a surge of attention due to equivalences to infinite-width neural networks (NNs) (Lee et al., 2018; Jacot et al., 2018). In this paper, we derive a simple theory of the generalization of KRR that yields estimators for many quantities of interest, including test risk and the covariance of the predicted function. Our framework is consistent with other recent works, such as those of Canatar et al. (2021) and Jacot et al. (2020), but is simpler and easier to derive. Our framework paints a new picture of KRR as an explicit competition between eigenmodes for a fixed budget of a quantity we term "learnability," and downstream generalization metrics can be expressed entirely in terms of the learnability received by each mode (Equations 7-14).

This picture stems from a *conservation law* latent in KRR which limits any kernel's ability to learn any complete basis of target functions. The conserved quantity, learnability, is the inner product of the target and predicted functions and, as we show, can be interpreted as a measure of how well the target function can be learned by a particular kernel given $n$ training examples. We prove that the total learnability, summed over a complete basis of target functions (such as the kernel eigenbasis), is no greater than the number of training samples, with equality at zero ridge parameter.

The conservation of this quantity suggests that it will prove useful for understanding the generalization of KRR. This intuition is borne out by our subsequent analysis: we derive a set of simple, closed-form estimates for test risk and other objects of interest and find that *all* of them can be transparently expressed in terms of eigenmode learnabilities. Our expressions are more compact and readily interpretable than those of prior work and constitute a major simplification.

Our derivation of these estimators is significantly simpler and more accessible than those of prior work, which relied on the heavy mathematical machinery of replica calculations and random matrix theory to obtain comparable results. By contrast, our approach requires only basic linear algebra, leveraging our conservation law at a critical juncture to bypass the need for advanced techniques.

We use our improved framework to shed light on several topics of interest:

    i) We provide a compelling theoretical explanation for the "deep bootstrap" phenomenon of Nakkiran et al. (2020) and identify two regimes of NN fitting occurring at early and late training times.

    ii) We generalize a previous result regarding the hardness of the parity problem for rotation-invariant kernels. Our technique is simple and illustrates the power of our framework.

    iii) We craft an estimator for predicted function *smoothness*, a new tool for the theoretical study of adversarial robustness.

    iv) We draw a tight analogy between our framework and the free Fermi gas, a well-studied statistical physics system, and thereby transfer insights into the free Fermi gas over to KRR.

We structure these applications as a series of vignettes.

The paper is organized as follows. We give preliminaries in Section 2. We define our conserved quantity and state its basic properties in Section 3. We characterize the generalization of KRR in terms of this quantity in Section 4. We check these results experimentally in Section 5. Section 6 consists of a series of short vignettes discussing topics (i)-(iv). We conclude in Section 7.

## 1.1 RELATED WORK

The present line of work has its origins with early studies of the generalization of Gaussian process regression (Opper, 1997; Sollich, 1999), with Sollich (2001) deriving an estimator giving the expected test risk of KRR in terms of the eigenvalues of the kernel operator and the eigendecomposition of the target function. We refer to this result as the "omniscient risk estimator[1]," as it assumes full knowledge of the data distribution and target function. Bordelon et al. (2020) and Canatar et al. (2021) brought these ideas into a modern context, deriving the omniscient risk estimator with a replica calculation and connecting it to the "neural tangent kernel" (NTK) theory of wide neural networks (Jacot et al., 2018), with (Loureiro et al., 2021) extending the result to arbitrary convex losses. Sollich & Halees (2002); Caponnetto & De Vito (2007); Spigler et al. (2020); Cui et al. (2021); Mallinar et al. (2022) study the asymptotic consistency and convergence rates of KRR in a similar vein. Jacot et al. (2020); Wei et al. (2022) used random matrix theory to derive a risk estimator requiring only training data. In parallel with work on KRR, Dobriban & Wager (2018); Wu & Xu (2020); Richards et al. (2021); Hastie et al. (2022) and Bartlett et al. (2021) developed equivalent results in the context of linear regression using tools from random matrix theory. In the present paper, we provide a new interpretation for this rich body of work in terms of explicit competition between eigenmodes, provide simplified derivations of the main results of this line of work, and break new ground with applications to new problems of interest. We compare selected works with ours and provide a dictionary between respective notations in Appendix A.

All prior works in this line — and indeed most works in machine learning theory more broadly — rely on approximations, asymptotics, or bounds to make any claims about generalization. The conservation law is unique in that it gives a sharp *equality* even at finite dataset size. This makes it a particularly robust starting point for the development of our framework (which does later make approximations).

In addition to those listed above, many works have investigated the spectral bias of neural networks in terms of both stopping time (Rahaman et al., 2019; Xu et al., 2019b;a; Xu, 2018; Cao et al., 2019; Su & Yang, 2019) and the number of samples (Valle-Perez et al., 2018; Yang & Salman, 2019; Arora et al., 2019). Our investigation into the deep bootstrap ties together these threads of work: we find that the interplay of these two sources of spectral bias is responsible for the deep bootstrap phenomenology.

## 2 PRELIMINARIES AND NOTATION

We study a standard supervised learning setting in which $n$ training samples $\mathcal{D} \equiv \{x_i\}_{i=1}^n$ are drawn i.i.d. from a distribution $p$ over $\mathbb{R}^d$. We wish to learn a (scalar) target function $f$ given noisy evaluations $\mathbf{y} \equiv (y_i)_{i=1}^n$ with $y_i = f(x_i) + \eta_i$, with $\eta_i \sim \mathcal{N}(0, \epsilon^2)$. As it simplifies later

---

[1] We borrow this terminology from Wei et al. (2022).

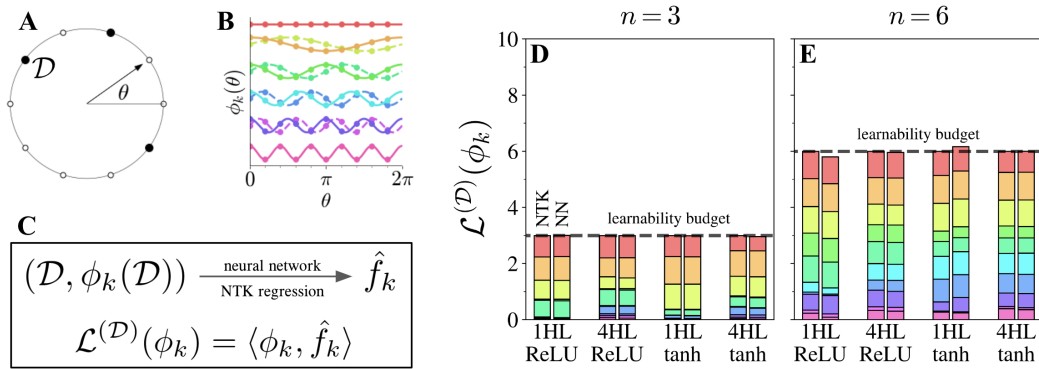

Figure 1: **Toy problem illustrating our conservation law.** **(A)** The task domain: the unit circle discretized into $M = 10$ points, $n$ of which comprise the dataset $\mathcal{D}$ (filled circles). **(B)** The 10 eigenfunctions of a rotation-invariant kernel on this domain, grouped into degenerate pairs and shifted vertically for clarity. **(C)** We use each eigenfunction $\phi_k$ in turn as the target function. For each $\phi_k$, we compute training targets $\phi_k(\mathcal{D})$, obtain a predicted function $\hat{f}_k$ in a standard supervised learning setup, and subsequently compute $\mathcal{D}$-learnability. This comprises 10 orthogonal learning problems. **(D,E)** Stacked bar charts with 10 components showing $\mathcal{D}$-learnability for each eigenfunction. The left bar in each pair contains results from NTK regression, while the right bar contains results from wide neural networks. Models vary in activation function and number of hidden layers (HL). Dashed lines indicate $n$. Learnabilities always sum to $n$, exactly for kernel regression and approximately for wide networks.

analysis, we assume this $\mathcal{N}(0, \epsilon^2)$ label noise is also applied to test targets[2]. Our results are easily generalized to vector-valued functions as in Canatar et al. (2021). For scalar functions $g, h$, we define $\langle g, h \rangle \equiv \mathbb{E}_{x \sim p}[g(x)h(x)]$ and $\|g\|^2 \equiv \langle g, g \rangle$.

We shall study the KRR predicted function $\hat{f}$ given by

$$\hat{f}(x) = \mathbf{k}_{x\mathcal{D}}(\mathbf{K}_{\mathcal{D}\mathcal{D}} + \delta \mathbf{I}_n)^{-1}\mathbf{y}, \tag{1}$$

where, for a positive-semidefinite kernel $K$, we have constructed the row vector $[\mathbf{k}_{x\mathcal{D}}]_i = K(x, x_i)$ and the empirical kernel matrix $[\mathbf{K}_{\mathcal{D}\mathcal{D}}]_{ij} = K(x_i, x_j)$ (which we trust to be nonsingular), $\delta$ is a ridge parameter, and $\mathbf{I}_n$ is the identity matrix. We wish to minimize test mean squared error (MSE) $\mathcal{E}^{(\mathcal{D})}(f) = \|f - \hat{f}\|^2 + \epsilon^2$ and its expectation over training sets $\mathcal{E}(f) = \mathbb{E}_{\mathcal{D}}[\mathcal{E}^{(\mathcal{D})}(f)]$ (where the expectation over $\mathcal{D}$ also averages over noise values). We emphasize that, here and in our discussion of learnability, $\hat{f}$ is understood to be the KRR predictor given by Equation 1 from training targets generated with target function $f$. In the classical bias-variance decomposition of test risk, we have bias $\mathcal{B}(f) = \|f - \mathbb{E}_{\mathcal{D}}[\hat{f}]\|^2 + \epsilon^2$ and variance $\mathcal{V}(f) = \mathcal{E}(f) - \mathcal{B}(f)$. We also define train MSE as $\mathcal{E}_{\text{tr}}(f) = \frac{1}{n}\sum_{i=1}^{n}(f(x_i) - \hat{f}(x_i))^2$.

## 2.1 THE KERNEL EIGENSYSTEM

By Mercer's theorem Mercer (1909), the kernel admits the decomposition $K(x, x') = \sum_i \lambda_i \phi_i(x)\phi_i(x')$, with eigenvalues $\lambda_i \geq 0$ and a basis of eigenfunctions $\phi_i$ satisfying $\langle \phi_i, \phi_j \rangle = \delta_{ij}$. We assume eigenvalues are indexed in descending order.

As the eigenfunctions form a complete basis, we are free to decompose $f$ and $\hat{f}$ as

$$f(x) = \sum_i \mathbf{v}_i \phi_i(x) \quad \text{and} \quad \hat{f}(x) = \sum_i \hat{\mathbf{v}}_i \phi_i(x), \tag{2}$$

where $\mathbf{v} \equiv (v_i)_i$ and $\hat{\mathbf{v}} \equiv (\hat{v}_i)_i$ are vectors of eigencoefficients.

---

[2]To study noiseless targets instead, simply subtract $\epsilon^2$ from expressions for test MSE.

## 3 LEARNABILITY AND ITS CONSERVATION LAW

Here we define learnability, our conserved quantity. Learnability is a measure of $\hat{f}$ defined similarly to test MSE, but it is linear instead of quadratic. For any function $f$ such that $\|f\| = 1$, let

$$\mathcal{L}^{(\mathcal{D})}(f) \equiv \langle f, \hat{f} \rangle \quad \text{and} \quad \mathcal{L}(f) \equiv \mathbb{E}_{\mathcal{D}}\left[\mathcal{L}^{(\mathcal{D})}(f)\right], \tag{3}$$

where, as with MSE, $\hat{f}$ is given by Equation 1. We refer to $\mathcal{L}^{(\mathcal{D})}(f)$ the $\mathcal{D}$-*learnability* of function $f$ with respect to the kernel and $n$ and refer to $\mathcal{L}(f)$ as the *learnability*. Up to normalization, this quantity is akin to the cosine similarity between $f$ and $\hat{f}$. We shall show that, for KRR, learnability gives a useful indication of how well a function (particularly a kernel eigenfunction) is learned. Results in this section are rigorous and exact; see Appendix G for proofs.

We begin by stating several basic properties of learnability to build intuition for the quantity.

**Proposition 3.1.** *The following properties of $\mathcal{L}^{(\mathcal{D})}$, $\mathcal{L}$, $\{\phi_i\}$, and any $f$ such that $\|f\| = 1$ hold:*

(a) $\mathcal{L}(\phi_i), \mathcal{L}^{(\mathcal{D})}(\phi_i) \in [0, 1]$.

(b) *When $n = 0$, $\mathcal{L}^{(\mathcal{D})}(f) = \mathcal{L}(f) = 0$.*

(c) *Let $\mathcal{D}_+$ be $\mathcal{D} \cup x$, where $x \in X, x \notin \mathcal{D}$ is a new data point. Then $\mathcal{L}^{(\mathcal{D}+)}(\phi_i) \geq \mathcal{L}^{(\mathcal{D})}(\phi_i)$.*

(d) $\frac{\partial}{\partial \lambda_i} \mathcal{L}^{(\mathcal{D})}(\phi_i) \geq 0$, $\frac{\partial}{\partial \lambda_i} \mathcal{L}^{(\mathcal{D})}(\phi_j) \leq 0$, *and* $\frac{\partial}{\partial \delta} \mathcal{L}^{(\mathcal{D})}(\phi_i) \leq 0$.

(e) $\mathcal{E}(f) \geq \mathcal{B}(f) \geq (1 - \mathcal{L}(f))^2$.

Properties (a-c) together give an intuitive picture of the learning process: the learnability of each eigenfunction monotonically increases from zero as the training set grows, attaining its maximum of one in the ridgeless, maximal-data limit. Properties (d) shows that the kernel eigenmodes are in competition — increasing one eigenvalue while fixing all others can only improve the learnability of the corresponding eigenfunction, but can only harm the learnabilities of all others — and that regularization only harms eigenfunction learnability. Property (e) gives a lower bound on MSE in terms of learnability and will be useful when we discuss the parity problem.

We now state the conservation law obeyed by learnability. This rule follows from the view of KRR as a projection of $f$ onto the $n$-dimensional subspace of the RKHS defined by the $n$ samples and is closely related to the "dimension bound" for linear learning rules given by Hsu (2021).

---

**Theorem 3.2** (Conservation of learnability). *For any complete basis of orthogonal functions $\mathcal{F}$, when ridge parameter $\delta = 0$,*

$$\sum_{f \in \mathcal{F}} \mathcal{L}^{(\mathcal{D})}(f) = \sum_{f \in \mathcal{F}} \mathcal{L}(f) = n, \tag{4}$$

*and when $\delta > 0$,*

$$\sum_{f \in \mathcal{F}} \mathcal{L}^{(\mathcal{D})}(f) < n \quad \text{and} \quad \sum_{f \in \mathcal{F}} \mathcal{L}(f) < n. \tag{5}$$

---

This result states that, summed over any complete basis of target functions, total learnability is at most the number of training examples, with equality at zero ridge[3]. This theorem is a stronger version of the classic "no-free-lunch" theorem for learning algorithms, which states that, averaged over *all* target functions, all models perform at chance level (Wolpert, 1996). While deep and compelling, this classic result is rarely informative in practice because the set of all possible target functions is prohibitively large to make a nonvacuous statement. By contrast, Theorem 3.2 requires only an average over a *basis* of target functions and, as we shall see, it is directly informative in understanding the generalization of KRR.

---

[3]We call Theorem 3.2 a *conservation law* because at zero ridge, as either the data distribution or the kernel changes, total learnability remains constant. This is analogous to, for example, physical conservation of charge.

# 4 THEORY

Eigenmode learnabilities are interpretable quantities, obeying a conservation law and several intuitive properties. These features suggest they may prove a wise choice of variables in a theory of KRR generalization. Here we show that this is indeed the case: we derive a suite of estimators for various metrics of KRR generalization, *all of which can be expressed entirely in terms of modewise learnabilities* and thereby inheriting their interpretability. Because they characterize KRR learning via eigenmode learnabilities, we call these equations the *eigenlearning equations*.

We sketch our method here and relegate the derivation to Appendix H. Our derivation leverages our conservation law to avoid the need for either a replica calculation or random matrix theoretic tools and is consequently more accessible and extensible than derivations of prior works. Results in this section are nonrigorous. Like comparable works, our derivations use an experimentally-validated "universality" assumption that the kernel features may be replaced by independent Gaussian features with the same statistics without changing downstream generalization metrics.

We begin by observing that $\hat{\mathbf{v}}$ depends linearly on $\mathbf{v}$, and we can thus construct a "learning transfer matrix" $\mathbf{T}^{(\mathcal{D})}$ such that $\hat{\mathbf{v}} = \mathbf{T}^{(\mathcal{D})}\mathbf{v}$. $\mathbf{T}^{(\mathcal{D})}$ is equivalent to the "KRR reconstruction operator" of Jacot et al. (2020) and, viewing KRR as linear regression in eigenfeature space, is essentially the "hat" matrix of linear regression. We then study $\mathbb{E}\left[\mathbf{T}^{(\mathcal{D})}\right]$, leveraging our universality assumption to show that it is diagonal with diagonal elements of the form $\lambda_i/(\lambda_i + \kappa)$, where $\kappa$ is a mode-independent constant. We show the mode-independence of $\kappa$ with an eigenmode-removal argument reminiscent of the cavity method of statistical physics (Del Ferraro et al., 2014). We use Theorem 3.2 to determine $\kappa$. Differentiating this result with respect to kernel eigenvalues, we obtain the covariance of $\mathbf{T}^{(\mathcal{D})}$ and thus of $\hat{\mathbf{v}}$, which permits evaluation of various test metrics[4].

## 4.1 THE EIGENLEARNING EQUATIONS

Let the effective regularization $\kappa$ be the unique positive solution to

$$n = \sum_i \frac{\lambda_i}{\lambda_i + \kappa} + \frac{\delta}{\kappa}. \tag{6}$$

We calculate various test and train metrics to be

(eigenmode learnability)
$$\mathcal{L}(\phi_i) = \mathcal{L}_i \equiv \frac{\lambda_i}{\lambda_i + \kappa}, \tag{7}$$

(overfitting coefficient)
$$\mathcal{E}_0 = n\frac{\partial \kappa}{\partial \delta} = \frac{n}{n - \sum_i \mathcal{L}_i^2}, \tag{8}$$

(test MSE)
$$\mathcal{E}(f) = \mathcal{E}_0 \left( \sum_i (1 - \mathcal{L}_i)^2 \mathbf{v}_i^2 + \epsilon^2 \right), \tag{9}$$

(bias of test MSE)
$$\mathcal{B}(f) = \sum_i (1 - \mathcal{L}_i)^2 \mathbf{v}_i^2 + \epsilon^2 = \frac{\mathcal{E}(f)}{\mathcal{E}_0}, \tag{10}$$

(variance of test MSE)
$$\mathcal{V}(f) = \mathcal{E}(f) - \mathcal{B}(f) = \frac{\mathcal{E}_0 - 1}{\mathcal{E}_0}\mathcal{E}(f), \tag{11}$$

(train MSE)
$$\mathcal{E}_{\mathrm{tr}}(f) = \frac{\delta^2}{n^2\kappa^2}\mathcal{E}(f), \tag{12}$$

(mean predictor)
$$\mathbb{E}[\hat{\mathbf{v}}_i] = \mathcal{L}_i\mathbf{v}_i, \tag{13}$$

(covariance of predictor)
$$\mathrm{Cov}[\hat{\mathbf{v}}_i, \hat{\mathbf{v}}_j] = \frac{\mathcal{E}(f)\mathcal{L}_i^2}{n}\delta_{ij}. \tag{14}$$

The learnability of an arbitrary normalized function can be computed as $\mathcal{L}(f) = \sum_i \mathcal{L}_i\mathbf{v}_i$. The mean of the target function evaluated at input $x$ can be obtained as $\mathbb{E}[\hat{f}(x)] = \sum_i \mathbb{E}[\mathbf{v}_i]\phi_i(x)$, with the covariance obtained similarly.

---

[4]For train risk, we simply quote the result of Canatar et al. (2021).

## 4.2 INTERPRETATION OF THE EIGENLEARNING EQUATIONS

Remarkably, all test metrics, and indeed all second-order statistics of $\hat{f}$, can be expressed solely in terms of modewise learnabilities, with no additional reference to eigenvalues required. This strongly suggests that we have identified the "correct" choice of variables for the problem. We now interpret these equations through the lens of learnability.

Equation 6 gives a constant $\kappa$ which decreases monotonically as $n$ increases. Equation 7 states that the learnability of eigenmode $i$ is 0 when $\kappa \gg \lambda_i$ and approaches 1 when $\kappa \ll \lambda_i$, in line with the observation of Jacot et al. (2020) that a mode is well-learned when $\kappa \ll \lambda_i$. Inserting Equation 7 into 6 and setting $\delta = 0$, we recover our conservation law.

Equation 9 is the omniscient risk estimator for test MSE. Modes with learnability equal to one are fully learned and do not contribute to the risk. Noise acts the same as target weight placed in modes with learnability zero. Equation 8 defines $\mathcal{E}_0$, the MSE when trained on pure-noise targets ($\mathbf{v}_i = 0$ and $\epsilon^2 = 1$). It is strictly greater than one and can be interpreted as the factor by which pure noise is overfit[5]. The denominator explodes when the $n$ units of learnability are fully allocated to the first $n$ modes, not distributed among a greater number ($\mathcal{L}_{i \leq n} = 1, \mathcal{L}_{i > n} = 0$). In this sense, overfitting of noise is "overconfidence" on the part of the kernel that the target function lies in the top-$n$ subspace, and "hedging" via a wider distribution of learnability (or sacrificing a portion of the learnability budget to the ridge parameter) lowers $\mathcal{E}_0$ and fixes this problem. This overconfidence occurs when the kernel eigenvalues drop sharply around index $n$ and is the cause of double-descent peaks (Belkin et al., 2019), which other works have also found can be avoided with an appropriate ridge parameter Canatar et al. (2021); Nakkiran et al. (2021).

Equations 10 and 11 show that, remarkably, the bias and variance can be expressed solely in terms of $\mathcal{E}(f)$ and $\mathcal{E}_0$. Since learnabilities strictly increase as $n$ grows, the bias strictly decreases while the variance can be nonmonotone, as also noted by Canatar et al. (2021). Equation 12 states that train error is related to test error by the target-independent proportionality constant $\delta^2/\kappa^2$. Finally, Equations 13 and 14 give the mean and covariance of the predicted eigencoefficients. Different eigencoefficients are uncorrelated, and the variance of $\hat{\mathbf{v}}_i$ is proportional to $\mathcal{L}_i^2$ but surprisingly independent of $\mathbf{v}_i$.

## 5 EXPERIMENTS

Here we describe experiments confirming our main results. The targets are eigenfunctions on three synthetic domains — the unit circle discretized into $M$ points, the (Boolean) hypercube $\{\pm 1\}^d$, and the $d$-sphere — as well as two-class subsets of MNIST and CIFAR-10. Unless otherwise stated, experiments use a fully-connected four-hidden-layer ReLU architecture, and finite networks have width 500. The kernel eigenvalues on each synthetic domain group into degenerate sets which we index by $k \in \mathbb{Z}^+$. Eigenvalues and eigencoefficients for image datasets are approximated numerically from a large sample of training data. Full experimental details can be found in Appendix B.

Figure 1 illustrates Theorem 3.2 in a toy setting. Modewise $\mathcal{D}$-learnabilities indeed sum to $n$. Figure 2 compares theoretical predictions for learnability and MSE with experiments on real and synthetic data, finding good agreement in all cases. Appendix C repeats one of these experiments with network widths varying from $\infty$ to 20, finding good agreement with theory even at narrow width.

## 6 VIGNETTES

### 6.1 EXPLAINING THE DEEP BOOTSTRAP

The *deep bootstrap* (DB) is a phenomenon observed by Nakkiran et al. (2020) in which the performance of a neural network stopped after a given number of training steps is relatively insensitive to the size of the training set, unless the training set is so small that it has been interpolated. The DB has been studied theoretically on kernel gradient flow (KGF), which describes the training of wide neural networks, by Ghosh et al. (2022) in the toy case in which the data lies on a high-dimensional

---

[5]See Mallinar et al. (2022) for a discussion of this interpretation.

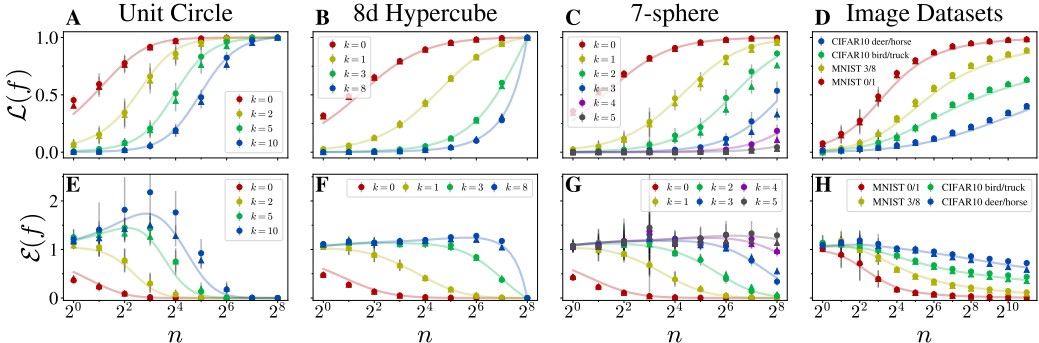

Figure 2: **Predicted learnabilities and MSEs closely match experiment. (A-D)** Learnability of various eigenfunctions on synthetic domains and binary functions over image datasets. Theoretical predictions from Equation 7 (curves) are plotted against experimental values from trained finite networks (circles) and NTK regression (triangles) with varying dataset size $n$. Error bars show one standard deviation of variation. **(E-H)** Same as (A-D) for test MSE, with theoretical predictions from Equation 9.

sphere. Here we give a convincing general explanation of this phenomenon using our framework and identify two regimes of NN fitting in the process.

Ali et al. (2019) proved that KRR with finite ridge generalizes remarkably similarly to KGF with finite stopping time (a theoretical result confirmed empirically by Lee et al. (2020) for NTKs on image datasets). When considering a standard supervised learning setting, the effective training time corresponding to a ridge $\delta$ is $\tau_{\text{eff}} \equiv \delta^{-1}n$ (see Appendix D for a discussion of this scaling). As a proxy for KGF, we shall study KRR as $\tau_{\text{eff}}$ increases from 0 to $\infty$.

In Equation 9, the ridge parameter affects MSE solely through the value of $\kappa$. Define $\kappa_0 \equiv \kappa|_{\delta=0}$, the minimum effective regularization at a given dataset size. In Appendix D, we show that, for powerlaw eigenspectra (as are commonly found in practice), there are two regimes of fitting:

1. **Regularization-limited regime**: $\tau_{\text{eff}} \ll \kappa_0^{-1}$ and $\kappa \approx \tau_{\text{eff}}^{-1}$. The generalization gap is small: $\mathcal{E}(f)/\mathcal{E}_{\text{tr}}(f) \approx 1$. Regularization dominates generalization, and adding training samples does not affect generalization[6].

2. **Data-limited regime**: $\tau_{\text{eff}} \gg \kappa_0^{-1}$ and $\kappa \approx \kappa_0$. The generalization gap is large: $\mathcal{E}(f)/\mathcal{E}_{\text{tr}}(f) \gg 1$. Data is interpolated, and decreasing regularization (i.e. increasing training time) does not affect generalization.

We suggest that Nakkiran et al. (2020) observe overlapping error curves for different $n$ at early times because, at these times, the model is in the regularization-limited regime.

We now present an experiment confirming our interpretation. In Figure 3, we reproduce an experiment of Nakkiran et al. (2020) illustrating the DB using ResNets trained on CIFAR-10, juxtaposing it with our proposed model for this phenomenon — KRR with varying $\tau_{\text{eff}}$ — trained on binarized MNIST. The match is excellent: in particular, both plots share the DB phenomenon that error curves for all $n$ overlap at early times, with test and train error peeling off the master curve at roughly the same time. We find that $\tau_{\text{eff}} \approx \kappa_0^{-1}$ is indeed when the transition between regimes occurs for KRR, matching our theoretical prediction. This experiment strongly suggests that both neural network and KRR fitting can be thought of as a transition between regularization-limited and data-limited regimes. See Appendix D for experimental details.

### 6.2 THE HARDNESS OF THE PARITY PROBLEM FOR ROTATION-INVARIANT KERNELS

The *parity problem* stands as a classic example of a function which is easy to write down but hard for common algorithms to learn. The parity problem was shown to be exponentially hard for Gaus-

---

[6]We mean here that adding training samples *while holding $\tau_{eff}$ constant* does not affect generalization.

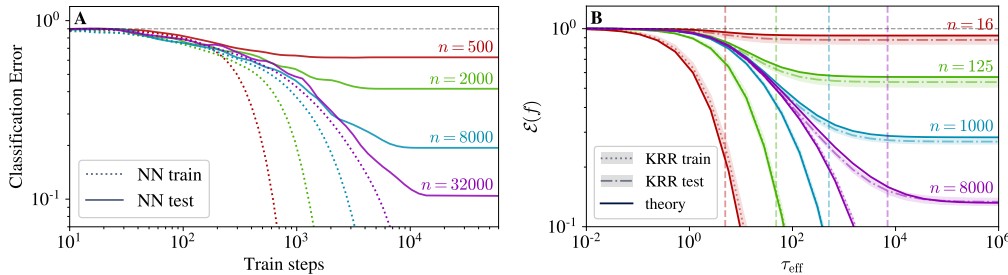

Figure 3: **We reproduce and explain the deep bootstrap phenomenon in KRR.** **(A)** An experiment illustrating the deep bootstrap effect using a ResNet-18 on CIFAR-10. **(B)** An analogous experiment using KRR on binarized MNIST. Eigenlearning predictions closely match experimental curves, and $\tau_{\text{eff}} = \kappa_0^{-1}$ (vertical dashed lines) faithfully predicts the transition from regularization-limited to data-limited fitting for each $n$.

sian kernel methods by Bengio et al. (2006). Here we generalize this result to KRR with arbitrary rotation-invariant kernels. Our analysis is made trivial by the use of our framework and is a good illustration of the power of working in terms of learnabilities.

The problem domain is the hypercube $\mathcal{X} = \{-1, +1\}^d$, over which we define the subset-parity functions $\phi_S(x) = (-1)^{\sum_{i \in S} \mathbb{1}[x_i=1]}$, where $S \subseteq \{1, ..., d\} \equiv [d]$. The objective is to learn $\phi_{[d]}$.

For any rotation-invariant kernel (such as the NTK of a fully-connected neural network), $\{\phi_S\}_S$ are the eigenfunctions over this domain, with degenerate eigenvalues $\{\lambda_k\}_{k=0}^d$ depending only on $k = |S|$. Yang & Salman (2019) proved that, for any fully-connected kernel, the even and odd eigenvalues each obey a particular ordering in $k$. Letting $d$ be odd for simplicity, this result and Equation 7 imply that $\mathcal{L}_1 \geq \mathcal{L}_3 \geq ... \geq \mathcal{L}_d$. Counting level degeneracies, this is a hierarchy of $2^{d-1}$ learnabilities of which $\mathcal{L}_d$ is the smallest. The conservation law of 3.2 then implies that $\mathcal{L}_d \leq \frac{n}{2^{d-1}}$, which, using Proposition 3.1(e), implies that

$$\mathcal{E}(\phi_{[d]}) \geq \left(1 - \frac{n}{2^{d-1}}\right)^2. \tag{15}$$

Obtaining an MSE below a desired threshold $\epsilon$ thus requires at least $n_{\min} = 2^{d-1}(1 - \epsilon^{1/2})$ samples, a sample complexity exponential in $d$. The parity problem is thus hard for all rotation-invariant kernels[7].

### 6.3 MEAN-SQUARED GRADIENT AND ADVERSARIAL ROBUSTNESS

While the omniscient risk estimate is the most important of the eigenlearning equations, we have also obtained estimators for arbitrary covariances of $\hat{f}$. Here we point out a first use for these covariances: studying the smoothness of $\hat{f}$ and thus its adversarial robustness. We fashion an estimator for function smoothness, confirm its accuracy with KRR experiments, and identify a discrepancy between intuition and experiment ripe for further exploration.

Consider the *mean squared gradient* (MSG) of $\hat{f}$ defined by $\mathcal{G}(\hat{f}) \equiv \mathbb{E}_x\left[|\nabla_x \hat{f}(x)|^2\right] = \|\nabla \hat{f}\|_2^2$. This quantity is a measure of function smoothness. Eigendecomposition yields that

$$\mathbb{E}_x\left[\left|\nabla_x \hat{f}(x)\right|^2\right] = \sum_{ij} \mathbb{E}[\hat{\mathbf{v}}_i \hat{\mathbf{v}}_j] \, g_{ij} \quad \text{with} \quad g_{ij} \equiv \mathbb{E}_x[\nabla_x \phi_i(x) \cdot \nabla_x \phi_j(x)]. \tag{16}$$

The expectation $\mathbb{E}[\hat{\mathbf{v}}_i \hat{\mathbf{v}}_j] = \mathbb{E}[\hat{\mathbf{v}}_i] \, \mathbb{E}[\hat{\mathbf{v}}_j] + \text{Cov}[\hat{\mathbf{v}}_i, \hat{\mathbf{v}}_j]$ is given by the eigenlearning equations, and the structure constants $g_{ij}$, which encode information about the domain, can be computed analytically for simple domains. On the $d$-sphere, for which the $\phi_i = \phi_{k\ell}$ are spherical harmonics, these are $g_{(k\ell),(k'\ell')} = k(k + d - 2)\delta_{kk'}\delta_{\ell\ell'}$.

---

[7]We note a number of recent approaches (Daniely & Malach, 2020; Kamath et al., 2020; Hsu, 2021) which give complexity lower bounds for learning parities using a simple degeneracy argument. However, these approaches do not leverage the spectral bias of the kernel and become vacuous when $k = d$.

Figure 4 shows the MSG of the function learned by KRR with a polynomial kernel trained on $k = 1$ modes on spheres of increasing dimension $d$, normalized by the MSG of the ground-truth target function. See Appendix E for experimental details and additional experiments in this vein. True MSG matches predicted MSG well in all settings, particularly at large $n$ and $d$.

The study of adversarial robustness currently suffers from a lack of theoretically tractable toy models, and we suggest our expression for MSG can help fill this gap. To illustrate this, we describe an insight that can be drawn from Figure 4. Vulnerability to gradient-based adversarial attacks can be viewed essentially as a phenomenon of surprisingly large gradients with respect to the input. If such vulnerability is an inevitable consequence of high dimension,

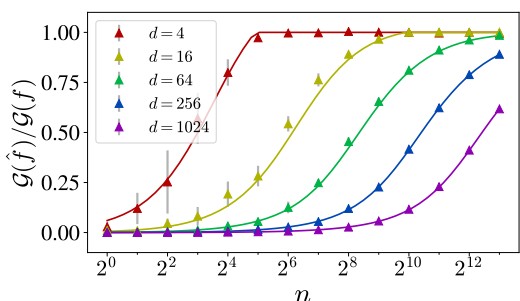

Figure 4: **Predicted function smoothness matches experiment.** Predicted MSG of $\hat{f}$ (curves) and empirical MSG for kernel regression (triangles) for $k = 1$ modes on hyperspheres with varying dimension.

a common heuristic belief (e.g. Gilmer et al. (2018)), one might expect that $\mathcal{G}(\hat{f})$ is generally much larger than $\mathcal{G}(f)$ at high dimension. Surprisingly, we see no such effect, a discrepancy which can be investigated further using our framework.

### 6.4 A QUANTUM MECHANICAL ANALOGY AND UNIVERSAL LEARNABILITY CURVES

Here we describe a remarkably tight analogy between our picture of KRR generalization and the statistics of the *free Fermi gas*, a canonical model in statistical physics. This allows certain insights into the free Fermi gas to be ported over to KRR. This correspondence is also of fundamental interest: KRR and the free Fermi gas are both paradigmatic systems in their respective fields, and it is remarkable that their statistics are in fact the same.

We defer the details of this correspondence to Appendix F and focus here on the takeaways. The free Fermi gas is defined by a scalar $\mu$ and a set of states with energies $\{\varepsilon_i\}_i$, each of which may be occupied or not. We find that $-\ln \kappa$ is analogous to $\mu$, the state energies are analogous to kernel eigenvalues, and the states' occupation probabilities are precisely analogous to the eigenmodes' learnabilities.

In all prior work and in our work thus far, the constant $\kappa$ has thus far been defined only as the solution to an *implicit* equation. Having an explicit equation would be advantageous. Leveraging methods for the study of the free Fermi gas, we find the following *explicit* formula for $\kappa$:

$$\kappa = \frac{m_n(\lambda_1, \lambda_2, ...)}{m_{n-1}(\lambda_1, \lambda_2, ...)}, \quad \text{where} \quad m_n(x_1, x_2, ...) \equiv \sum_{1 \leq j_1 < ... < j_n} x_{j_1}...x_{j_n}. \quad (17)$$

The second takeaway is the identification of a universal behavior in the learning dynamics of KRR. In systems obeying Fermi-Dirac statistics, a plot of $p_i$ vs. $\varepsilon_i$ takes a characteristic sigmoidal shape. As shown in Figure 10, we robustly see this sigmoidal shape in plots of $\mathcal{L}_i$ vs. $\ln \lambda_i$. As the number of samples and even the task are varied, this sigmoidal shape remains universal, merely translating horizontally (in particular, moving left as samples are added). This sigmoidal shape is thus a signature of KRR and could in principle be used to, for example, determine if an unknown kernel method resembles KRR.

## 7 CONCLUSIONS

We have developed an interpretable unified framework for understanding the generalization of KRR centered on a previously-unexploited conserved quantity. We then used this improved framework to break new theoretical ground in a variety of subjects including the parity problem, the deep bootstrap, and adversarial robustness and developed a tight analogy between KRR and canonical physical system allowing the transfer of insights. We have covered much territory, and each of these subjects is ripe for further exploration in future work.

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

## A    NOTATION DICTIONARY AND COMPARISON WITH RELATED WORKS

Tables 1, 2 and 3 provide a dictionary between the notations of our paper and the nearest related works. Rows should be read in comparison with the top row and interpreted as "when the present paper writes X, the other paper would write Y for the same quantity." Comparing expressions for test MSE in Table 2 and predicted function covariance in Table 3 makes it very clear that our learnability framework permits simpler and much more interpretable expression of these results than previously given.

Table 1: Notation dictionary between the present paper and related works (part 1).

| Paper | # samples | ridge | noise | eigenvalues | eigenfns | eigencoeffs |
|---|---|---|---|---|---|---|
| (ours) | $n$ | $\delta$ | $\epsilon^2$ | $\lambda_i$ | $\phi_i$ | $\mathbf{v}_i$ |
| Bordelon et al. (2020) | $p$ | $\lambda$ | NA | $\lambda_\rho$ | $\lambda_\rho^{-1/2}\psi_\rho$ | $\lambda_\rho^{1/2}\bar{w}_\rho$ |
| Canatar et al. (2021) | $P$ | $\lambda$ | $\sigma^2$ | $\eta_\rho$ | $\eta_\rho^{-1/2}\psi_\rho$ | $\eta_\rho^{1/2}\bar{w}_\rho$ |
| Jacot et al. (2020) | $N$ | $N\lambda$ | $\epsilon^2$ | $d_k$ | $f^{(k)}$ | $\langle f^{(k)}, f^* \rangle$ |
| Cui et al. (2021)[8] | $n$ | $n\lambda$ | $\sigma^2$ | $\eta_k$ | $\phi_k$ | $\eta_k^{1/2}\vartheta_k^*$ |

Table 2: Notation dictionary between the present paper and related works (part 2).

| Paper | eff. reg. | overfitting coeff. | test MSE |
|---|---|---|---|
| (ours) | $\kappa$ | $\mathcal{E}_0$ | $\mathcal{E}(f) = \mathcal{E}_0\left(\sum_i (1-\mathcal{L}_i)^2 \mathbf{v}_i^2 + \epsilon^2\right)$ |
| Bordelon et al. (2020) | $\frac{t+\lambda}{n}$ | $\left(1 - \frac{p\gamma}{(t+\lambda)^2}\right)^{-1}$ | $E_g = \sum_\rho \frac{\bar{w}_\rho^2}{\lambda_\rho}\left(\frac{1}{\lambda_\rho} + \frac{p}{\lambda+t}\right)^{-2}\left(1 - \frac{p\gamma}{(\lambda+t)^2}\right)^{-1}$ |
| Canatar et al. (2021) | $\frac{\kappa}{n}$ | $\frac{1}{1-\gamma}$ | $E_g = \frac{1}{1-\gamma}\sum_\rho \frac{\eta_\rho}{(\kappa+P\eta_\rho)^2}\left(\kappa^2\bar{w}_\rho^2 + \sigma^2 P\eta_\rho\right)$ |
| Jacot et al. (2020) | $\vartheta$ | $\partial_\lambda \vartheta$ | $\tilde{R}^\epsilon = \partial_\lambda\vartheta\left(\left\|(I_c - \tilde{A}_\vartheta)f^*\right\|^2 + \epsilon^2\right)$ |
| Cui et al. (2021)[9] | $\frac{z}{n}$ | $\dfrac{1}{1 - \frac{1}{n}\sum\limits_{k=1}^{P}\frac{\eta_k^2}{(z/n+\eta_k)^2}}$ | $\epsilon_g = \dfrac{\frac{z^2}{n^2}\sum\limits_{k=1}^{\infty}\frac{\vartheta_k^{*2}\eta_k}{(z/n+\eta_k)^2}+\sigma^2}{1 - \frac{1}{n}\sum\limits_{k=1}^{\infty}\frac{\eta_k^2}{(z/n+\eta_k)^2}}$ |

---

[8]Cui et al. (2021) use simplifications of the expressions of Loureiro et al. (2021), whose notation does not map cleanly onto our tables.

[9]The overfitting coefficient and test MSE estimator of Cui et al. (2021) appear more complex than the alternatives in part because other works define intermediate quantities like $\mathcal{E}_0$, $\gamma$, etc., which include sums over eigenmodes.

Table 3: Notation dictionary between the present paper and related works (part 3).

| Paper | predicted fn covariance |
|---|---|
| (ours) | $\mathrm{Cov}[\hat{\mathbf{v}}_i, \hat{\mathbf{v}}_j] = \frac{\mathcal{L}_i^2 \mathcal{E}(f)}{n}\,\delta_{ij}$ |
| Bordelon et al. (2020) | NA |
| Canatar et al. (2021) | $\sqrt{\eta_\alpha \eta_\beta}\,\mathrm{Cov}_{\mathcal{D}}\left[w_\alpha^*, w_\beta^*\right] = \frac{1}{1-\gamma}\left(\sigma^2 + \kappa^2 \sum_\rho \frac{\eta_\rho \bar{w}_\rho^2}{(P\eta_\rho + \kappa)^2}\right)\frac{P\eta_\alpha^2}{(P\eta_\alpha + \kappa)^2}\delta_{\alpha\beta}$ |
| Jacot et al. (2020)[10] | $V_k(f^*, \lambda, N, \epsilon) = \frac{\partial_\lambda \vartheta(\lambda)}{N}\left(\left\|\left(I_C - \tilde{A}_\vartheta\right)f^*\right\|_S^2 + \epsilon^2 + \left\langle f^{(k)}, f^*\right\rangle_S^2 \frac{\vartheta^2(\lambda)}{(\vartheta(\lambda) + d_k)^2}\right)\frac{d_k^2}{(\vartheta(\lambda) + d_k)^2}$ |
| Cui et al. (2021) | NA |

Sollich (2001) derived the omniscient risk estimate in the context of GP regression and was, to our knowledge, the first to obtain the result. As its notation does not map cleanly onto the variables we use, we do not include this work in the above tables. Other analogous works include Wu & Xu (2020), Richards et al. (2021), Hastie et al. (2022) (see Definition 1) and Bartlett et al. (2021) (see Theorem 4.13), which derive comparable equations for the test MSE of linear ridge regression (LRR), which, taking the dual view of KRR as LRR in the kernel embedding space, is essentially the same problem. All comparable works involve solving a self-consistent equation for an effective regularization parameter, which we call $\kappa$ in our work.

It is worth pointing out that this literature is largely divided into two parallel camps: the works in the tables above, together with Loureiro et al. (2021) and Cohen et al. (2021) (who study GP regression using field theoretic tools), study the generalization of KRR, largely using approaches from statistical physics, while Dobriban & Wager (2018); Wu & Xu (2020); Richards et al. (2021); Hastie et al. (2022); Bartlett et al. (2021) study LRR using random matrix theory. Despite studying virtually the same problem, these camps have largely developed unaware of each other, and we hope the present paper helps unify them. The KRR works all rely on approximations, particularly the spectral "universality" approximation that the eigenfunctions are random and structureless and the design matrix can thus be replaced by a random Gaussian matrix. The works of Bordelon et al. (2020); Canatar et al. (2021), as well as ours, make additional approximations valid for reasonable eigenspectra and large $n$, while Jacot et al. (2020) does not make approximations and provides bounds instead (though these bounds diverge at zero ridge, highlighting the difficulty of studying interpolating methods). On the other side, the LRR works do not make a unversality approximation, but instead assume in their setting that the feature vectors are high-dimensional and have (sub)Gaussian moments (or some similar condition), which ultimately amounts to the same condition, now enforced in the setting instead of assumed as an approximation. These works are typically mathematically rigorous. Relative to the LRR works, one contribution of the KRR works is the empirical observation that this universality approximation is generally accurate in practice[11], even for NTKs at initialization (for NTKs after training, the story appears mixed; see Wei et al. (2022) for evidence for universality and Ba et al. (2022) for evidence against it).

While our work is generally consistent with other works in the KRR set, we note one discrepancy regarding the covariance of the predicted function. Our expression, Equation 14, agrees with that of Canatar et al. (2021)[12]. However, the expression of Jacot et al. (2020), given in Theorem 2, contains an extra $\mathcal{O}(n^{-1})$ term (blue in Table 3) dependent upon the target eigencoefficient of the mode in question. This term is small enough that it can be removed without affecting the bounds the authors prove. It contributes negligibly when summing over all eigenmodes to compute e.g. test

---

[11]This is a nontrivial observation, as is not hard to construct contrived cases in which this approximation does *not* hold, as discussed by Sollich & Halees (2002). The conditions under which this universality assumption holds are an active area of research. A variety of rigorous works have proven this spectral universality assumption in specific high-dimensional settings (El Karoui, 2010; Cheng & Singer, 2013; Fan & Montanari, 2019; Liu et al., 2021; Lu & Yau, 2022), and Tomasini et al. (2022) has shown it does not quite hold in certain noisy low-dimensional settings.

[12]Equation 71 in the supplement of their published paper, giving the covariance of the predicted function, contains an spurious extra term, but this has been fixed in the arXiv version.

error or mean squared gradient, but contributes to leading order when interrogating the variance of a particular mode. A sanity-check calculation that the sum of variance over all modes ought to equal $\mathcal{V}(f)$ provides evidence that our equations, which lack this term, are correct.

Lastly, we note that Cohen et al. (2021) study Gaussian process regression in the spirit of Sollich (1999) using field theoretic tools. Though they do not explicitly discuss the omniscient risk estimate, Canatar et al. (2021) note that their "equivalence kernel + corrections" learning curve can be viewed as a perturbative expansion of the omniscient risk estimate.

## B  EXPERIMENTAL METHODS

### B.1  SYNTHETIC DOMAINS

Our experiments use both real image datasets and synthetic target functions on the following three domains:

1. *Discretized Unit Circle.*  We discretize the unit circle into $M$ points, $\mathcal{X} = \{(\cos(2\pi j/M), \sin(2\pi j/M)\}_{j=1}^{M}$. Unless otherwise stated, we use $M = 256$. The eigenfunctions on this domain are $\phi_0(\theta) = 1$, $\phi_k(\theta) = \sqrt{2}\cos(k\theta)$, and $\phi_k'(\theta) = \sqrt{2}\sin(k\theta)$, for $k \geq 1$.

2. *Hypercube.*  We use the vertices of the $d$-dimensional hypercube $\mathcal{X} = \{-1, 1\}^d$, giving $M = 2^d$. As stated in Section 6.2, the eigenfunctions on this domain are the subset-parity functions with eigenvalues determined by the number of sensitive bits $k$.

3. *Hypersphere.*  To demonstrate that our results extend to continuous domains, we perform experiments on the $d$-sphere $\mathbb{S}^d \equiv \{x \in \mathbb{R}^{d+1} | x^2 = 1\}$. The eigenfunctions on this domain are the hyperspherical harmonics (see, e.g., Frye & Efthimiou (2012); Bordelon et al. (2020)) which group into degenerate sets indexed by $k \in \mathbb{N}$. The corresponding eigenvalues decrease exponentially with $k$, and so when summing over all eigenmodes when computing predictions, we simply truncate the sum at $k_{\max} = 70$.

Eigenvalues and multiplicities for these synthetic domains are shown in Figure 5.

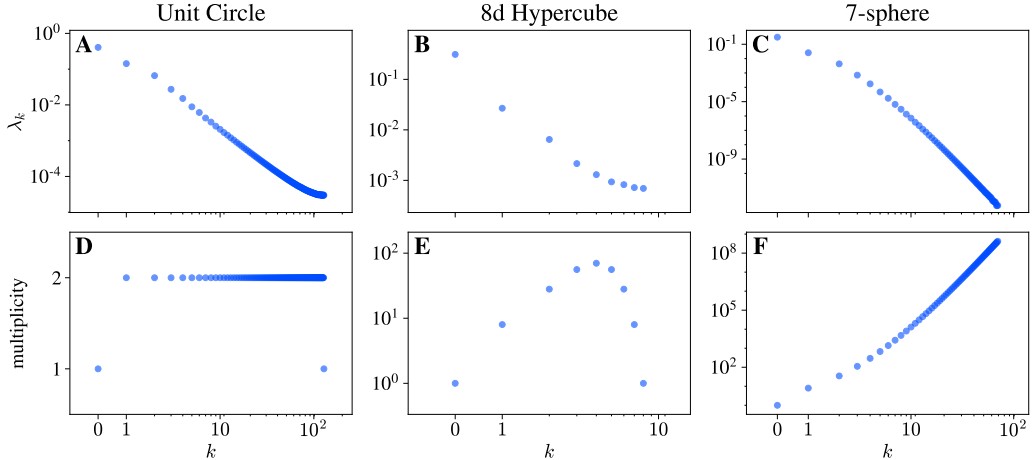

Figure 5: **4HL ReLU NTK eigenvalues and multiplicities on three synthetic domains.** (A) Eigenvalues for $k$ for the discretized unit circle ($M = 256$). Eigenvalues decrease as $k$ increases except for a few near exceptions at high $k$. (B) Eigenvalues for the 8d hypercube. Eigenvalues decrease monotonically with $k$. (C) Eigenvalues for the 7-sphere up to $k = 70$. Eigenvalues decrease monotonically with $k$. (D) Eigenvalue multiplicity for the discretized unit circle. All eigenvalues are doubly degenerate (due to $cos$ and $sin$ modes) except for $k = 0$ and $k = 128$. (E) Eigenvalue multiplicity for the 8d hypercube. (F) Eigenvalue multiplicity for the 7-sphere.

### B.2 IMAGE DATASET EIGENINFORMATION

Experiments with binary image classification tasks used scalar targets of $\pm 1$. To obtain the eigeninformation necessary for theoretical predictions, we select a large training set with $M \sim \mathcal{O}(10^4)$ samples, computing the eigensystem of the data-data kernel matrix to obtain $M$ eigenvalues $\{\lambda_i\}_i$ and target vector eigencoefficients $\mathbf{v}$. We then compute theoretical learnability and MSE as normal.

The resulting test MSE predictions obtained this way correspond to an average over the $M$ points in the sample, $n$ of which are in fact part of the training set (compare with the discrete and continuous settings of Appendix H.1, and thus they are too low when $n/M$ is not negligible. We wish to predict purely the *off-training-set* error $\mathcal{E}_{\text{OTS}}$. We do so by noting that $M\mathcal{E} = n\mathcal{E}_{\text{tr}} + (M - n)\mathcal{E}_{\text{OTS}}$ and solving for $\mathcal{E}_{\text{OTS}}$ (doing the same thing for for learnability). Another solution to this problem is to change the experimental design so train and test data are sampled from the same large pool (as done by Bordelon et al. (2020) and Canatar et al. (2021)), but this trick allows us a more standard setup in which train and test data are disjoint. Experimental test metrics for image datasets are computed with a random subset of $M' \sim \mathcal{O}(1k)$ images from the test set.

Eigenvalues and multiplicities for four two-class image datasets are shown in Figure 6.

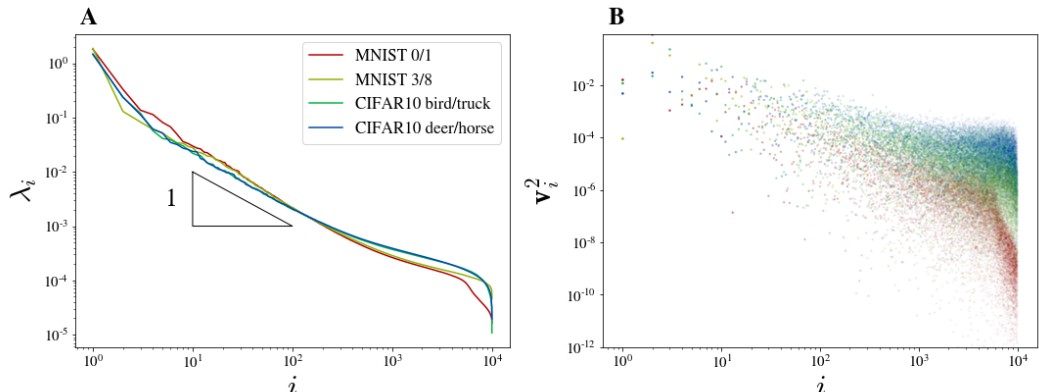

Figure 6: **Eigenvalues and eigencoefficients for four binary image classification tasks.**
**(A)** Kernel eigenvalues as computed from $10^4$ training points as described in Appendix B. Spectra for CIFAR10 tasks roughly follow power laws with exponent $-1$, while spectra for MNIST tasks follow power laws with slightly steeper descent. **(B)** Eigencoefficients as computed from $10^4$ training points. Tasks with higher observed learnability (Figure 2) place more weight in higher (i.e., lower-index) eigenmodes and less in lower ones.

### B.3 RUNTIMES

For the dataset sizes we consider in this paper, exact NTK regression is typically quite fast, running in seconds, while the training time of finite networks varies from seconds to minutes and depends on width, depth, training set size, and eigenmode. In particular, as described by Rahaman et al. (2019), lower eigenmodes take longer to train (especially when aiming for near-zero training MSE as we do here).

### B.4 HYPERPARAMETERS

We conduct all our experiments using JAX (Bradbury et al., 2018), performing exact NTK regression with the neural_tangents library (Novak et al., 2019) built atop it. Unless otherwise stated, all experiments used four-hidden-layer ReLU networks initialized with NTK parameterization (Sohl-Dickstein et al., 2020) with $\sigma_w = 1.4$, $\sigma_b = .1$. The tanh networks used in generating Figure 1 instead used $\sigma_w = 1.5$. Experiments on the unit circle always used a learning rate of .5, while experiments on the hypercube, hypersphere, and image datasets used a learning rate of .5 or .1 depending on the experiment. While higher learning rates led to faster convergence, they tended to match theory more poorly, in line with the large learning rate regimes described by Lewkowycz

et al. (2020). Means and one-standard-deviation error bars always the statistics from several random dataset draws and initializations (for finite nets):

- The toy experiment of Figure 1 used only a single trial per eigenmode by design.
- Other experiments on synthetic domains used 30 trials.
- The experiments on image datasets in Figure 2 used 15 trials.

### B.5 INITIALIZING THE NETWORK FUNCTION AT ZERO

Naively, when training an infinitely-wide network, the NTK only describes the *mean* learned function, and the true learned function will include an NNGP-kernel-dependent fluctuation term reflecting the random initialization (Lee et al., 2019). However, by storing a copy of the parameters at $t = 0$ and redefining $\hat{f}_t(x) := \hat{f}_t(x) - \hat{f}_0(x)$ throughout optimization and at test time, this term becomes zero. We use this trick in our experiments with finite networks.

## C  VARYING WIDTH EXPERIMENT

While kernel regression describes only infinitely-wide neural networks, it is natural to wonder whether our equations are nonetheless informative outside this regime. Figure 7 compares predicted learnability and MSE with experiment for hypercube eigenmodes with networks with varying widths. Our predictions remain informative down to width 50, and lower-frequency modes are better learned at all widths, suggesting that kernel eigenanalysis has a role to play even in the study of feature learning.

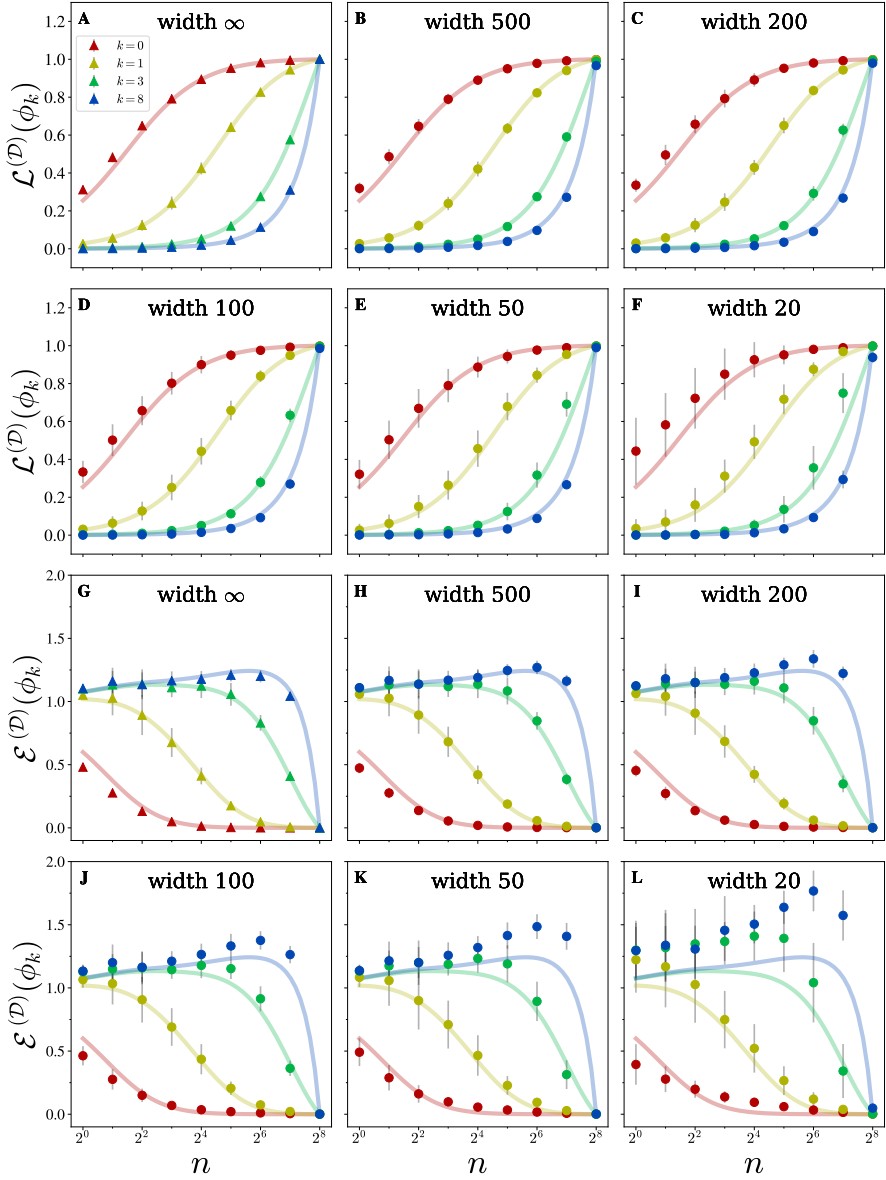

Figure 7: **Comparison between predicted learnability and MSE for networks of various widths. (A-F)** Predicted (curves) and true (triangles and circles) learnability for four eigenmodes on the 8d hypercube. Dataset size $n$ varies within each subplot, and the width of the 4HL ReLU network varies between subplots. **(G-L)** Same as (A-F) but with MSE instead of learnability.

## D    EXPLAINING THE DEEP BOOTSTRAP IN KRR

In this appendix, we use the eigenlearning equations to arrive at an explanation of the deep bootstrap phenomenon of Nakkiran et al. (2020). As explained in the main text, we study KRR as a proxy for KGF. We begin by finding the relationship between the ridge parameter and the effective training time (we assume for simplicity that the learning rate of KGF is one). Ali et al. (2019) obtained their result regarding the similarity of KRR and KGF[13] under the identifcation [time] = [ridge]$^{-1}$. They assume a different ridge scaling than ours, and, in our notation, we identify $\tau_{\text{eff}} = \delta^{-1}n$. Their KGF scaling is correct for modeling a standard supervised learning setup, as can be verified by taking the KGF limit of ordinary minibatch SGD.

To study the effect of this "early stopping" on generalization, we must find how finite $\tau_{\text{eff}}$ affects the constant $\kappa$. Recalling Equation 6 defining $\kappa$,

$$n = \sum_i \frac{\lambda_i}{\lambda_i + \kappa} + \frac{\delta}{\kappa}, \tag{6}$$

we can easily see that $\kappa \leq (n^{-1}\sum_i \lambda_i + \tau_{\text{eff}}^{-1})$ and $\kappa \geq \tau_{\text{eff}}^{-1})$. Squeezing $\kappa$ with these bounds, we find that, when $n \gg \tau_{\text{eff}}^{-1}\sum_i \lambda_i$, then $\kappa \approx \tau_{\text{eff}}^{-1}$. For any $\tau_{\text{eff}}$, there thus exists some $n$ above which additional data does not further lower $\kappa$ and permit the learning of additional eigenmodes. Let us call this the *regularization-limited regime*, and its counterpart (in which $\tau_{\text{eff}}$ is sufficiently large and regularization is effectively zero) the *data-limited regime*.

We can find the crossover regularization provided knowledge of the eigenspectrum. Let us assume that $\lambda_i \sim i^{-\alpha}$ for some $\alpha > 1$. Mallinar et al. (2022) show that $\kappa_0 \equiv \kappa|_{\delta=0} \to \left(\alpha^{-1}\pi\csc(\alpha^{-1}\pi)\right)^\alpha n^{-\alpha}$ as $n \to \infty$. Extending their analysis to finite ridge, we find that, at large $n$, $n = n(\kappa/\kappa_0)^{-1/\alpha} + (\delta\kappa)^{-1}$, or equivalently

$$1 = \left(\frac{\kappa_0}{\kappa}\right)^{\frac{1}{\alpha}} + \frac{1}{\tau_{\text{eff}}\kappa}. \tag{18}$$

Inspection of Equation 18 reveals that when $\tau_{\text{eff}} \ll \kappa_0^{-1}$, then $\kappa \approx \tau_{\text{eff}}^{-1}$, and when $\tau_{\text{eff}} \gg \kappa_0^{-1}$, then $\kappa \approx \kappa_0$. We thus expect a crossover from the regularization-limited to the data-limited regime when $\tau_{\text{eff}} \approx \kappa_0$.

In Figure 8, we plot the theoretical omniscient risk estimate using powerlaw eigenvalues ($\lambda_i \sim i^{-\alpha}$) and powerlaw eigencoefficients $\mathbf{v}_i^2 \sim i^{-\alpha}$) with $\alpha = 2$ for various $n$. The close resemblance of these curves to the experimental KRR curves of Figure 3 confirm that our powerlaw toy model is good. If desired, Figures 3(A,B) and Figure 8 can be viewed as the distillation of the DB phenomenon in three stages of increasing artificiality.

It is worth noting that increasing $n$ serves to both decrease $\kappa$ and decrease $\mathcal{E}_0$. For powerlaw spectra, as $n$ increases with fixed $\tau_{\text{eff}}$, $\mathcal{E}_0$ tends to its lower bound of 1. It approaches 1 when the regularization-limited regime begins.

Similar ideas regarding the scaling of KRR are discussed by Cui et al. (2021). The notion of regimes limited by either dataset size or training time is explored using the language of scaling laws by Kaplan et al. (2020) and Bahri et al. (2021). We expect that this analysis could be extended to proper KGF using the analysis of Bordelon & Pehlevan (2021).

### D.1    DEEP BOOTSTRAP EXPERIMENTS

In replicating the deep bootstrap phenomenon for neural networks, we use the same experimental setup as Nakkiran et al. (2020), a ResNet-18 architecture trained on CIFAR-10 with data augmentation (random horizontal flips and random crops). We optimize cross-entropy loss using SGD, with batchsize 128, momentum 0.9, and initial learning rate 0.1 with cosine decay. The test/train curves are the mean of 4 training runs; the curves have been gaussian-smoothed to remove high-frequency noise artifacts.

---

[13]More precisely, they study *linear* ridge regression and *linear* gradient flow, but their result also applies to the kernelized versions of these algorithms.

We perform kernel ridge regression (KRR) using the NTK of a fully-connected network with four hidden layers of width 500. We binarize MNIST into two classes: $\{0, 1, 2, 3, 4\}$ and $\{5, 6, 7, 8, 9\}$. Although the test curves shift slightly depending on the binarization scheme, the theory curves all fall within one standard deviation of the empirical curves for all binarizations we tried. The KRR test/train curves are the mean of 10 training runs.

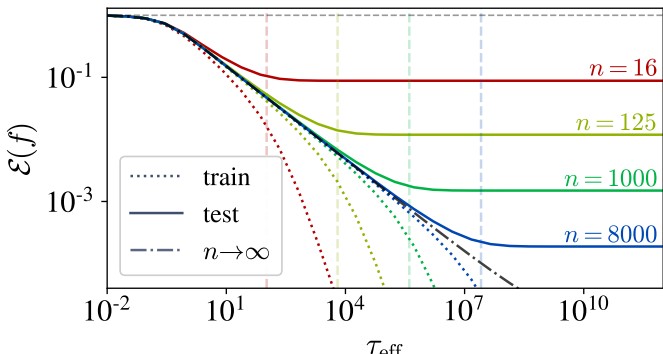

Figure 8: **Deep bootstrap theoretical test/train curves for a synthetic kernel spectrum.** Here we illustrate the deep bootstrap phenomenon using a hypothetical kernel giving powerlaw eigenvalues and eigencoefficients with exponent $\alpha = 2$. In this setting, we note that finite-data test/train curves simultaneously split from the $n \to \infty$ "online learning curve" (black dot-dashed line). We see that $\tau_{\text{eff}} = \kappa_0^{-1}$ (vertical dashed lines) predicts the transition from regularization-limited to data-limited fitting for each $n$. (We choose $\alpha = 2$ for a clean illustration of the phenomenon; empirically, CIFAR-10 and MNIST give spectra with exponents closer to 1.)

## E    ADDITIONAL MEAN-SQUARED GRADIENT EXPERIMENTS

In the MSG experiment of 4, we run KRR with a polynomial kernel on data sampled from hyperspheres of varying dimension. The polynomial kernel is $K(x, x') = 1 + x^\top x' + (x^\top x')^2 + (x^\top x')^3$. We perform this experiment using PyTorch (Paszke et al., 2019) and compute $\nabla \hat{f}$ numerically.

In the MSG experiment of 9, we train finite-width FCNs on eigenmodes on the (discretized) unit circle.

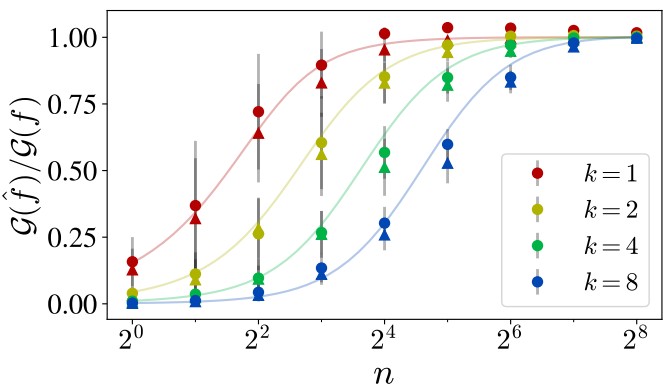

Figure 9: **Predicted function smoothness matches experiment on the unit circle.** Theoretical MSG predictions (curves) and empirical values for finite networks (circles) and kernel regression (triangles) for various eigenmodes on the discretized unit circle with $M = 256$, normalized by the ground-truth mean squared gradient of $\mathcal{G}(f) = \mathbb{E}\big[|f'(x)|^2\big] = k^2$. Because this is a discrete domain, smoothness is computed using a discretization as $\mathcal{G}(\hat{f}) = \mathbb{E}_j\Big[|\hat{f}(x_j) - \hat{f}(x_{j+1})|^2\Big]$, where $x_j$ and $x_{j+1}$ are neighboring points on the unit circle.

## F  QUANTUM MECHANICAL ANALOGY

### F.1  EXPLICIT EQUATION FOR $\kappa$

Here we develop a tight analogy between our picture of KRR and the *free Fermi gas*. The free Fermi gas consists of a collection of single-particle orbitals with energies $\{\varepsilon_i\}_i$ connected to a bath of particles at chemical potential $\mu$. In our analogy, we will identify orbital $i$ with kernel eigenmode $i$. Each orbital will contain zero or one fermions ($n_i \in \{0, 1\}$), with the occupation probability of orbital $i$ given by the Fermi-Dirac distribution to be $\langle n_i \rangle = (1 + e^{\varepsilon_i - \mu})^{-1}$. If we identify $\varepsilon_i = -\ln \lambda_i$ and $\mu = -\ln \kappa$, we find that $\langle n_i \rangle = \mathcal{L}_i$: the orbital occupation probability is precisely the eigenmode learnability.

We can always choose $\kappa$ so that eigenmode learnabilities sum to $n$. This is equivalent to the statement that we can always choose the chemical potential $\mu$ to ensure the system contains $n$ fermions on average. An eigenmode with eigenvalue $\lambda_i \geq \kappa$ receives at least half a unit of learnability, and an orbital with energy $\varepsilon_i \leq \mu$ is occupied with probability at least one half.

Here we elaborate further the physical analogy laid out in Section 6.4. In our system of noninteracting fermions, we have thus far identified

$$-\ln \lambda_i \Leftrightarrow \varepsilon_i \text{ (energy of orbital } i) \tag{19}$$

$$-\ln \kappa \Leftrightarrow \mu \text{ (chemical potential)} \tag{20}$$

$$\mathcal{L}_i \Leftrightarrow \langle n_i \rangle \text{ (expected occupancy of orbital } i). \tag{21}$$

What is the value of $\kappa$? Observe that

$$\langle n_i \rangle = \frac{1}{1 + e^{\varepsilon_i - \mu}} \tag{22}$$

gives the expected occupation number *in the grand canonical ensemble* where the total number of fermions is allowed to fluctuate. By the equivalence of thermodynamic ensembles, we expect to get the same answer for $\langle n_i \rangle$ in the *canonical ensemble* where the total number of fermions does not fluctuate and is exactly $n$. This suggests that we should attempt to compute $\langle n_i \rangle$ in the canonical ensemble. By comparing the answer to the grand canonical expression, we can solve for $\mu$ and hence for $\kappa$.

We assume a total number of orbitals $M \gg n$ (which may be infinite). Note that the equivalence of ensembles holds only in the thermodynamic limit $n \gg 1$, so we must take this limit at the end.

In the canonical ensemble, each microstate is labeled by of a list of indices $1 \leq j_1 < ... < j_n \leq M$ corresponding to the occupied orbitals. Direct computation of the canonical partition function shows that

$$Z_\mathrm{C} = m_n(e^{-\varepsilon_1}, ...., e^{-\varepsilon_D}), \tag{23}$$

where $m_n$ is the so-called "elementary symmetric polynomial of order $n$":

$$m_n(x_1, ..., x_M) = \sum_{1 \leq j_1 < ... < j_n \leq M} x_{j_1}...x_{j_n}. \tag{24}$$

We also define the same with index $i$ disallowed:

$$m_n^{(i)}(x_1, ..., x_M) = \sum_{\substack{1 \leq j_1 < ... < j_n \leq M \\ j_k \neq i \ \forall \ k}} x_{j_1}...x_{j_n}. \tag{25}$$

We find that

$$\langle n_i \rangle = -\frac{\partial \ln Z_\mathrm{C}}{\partial \varepsilon_i} = e^{-\varepsilon_i} \frac{m_{n-1}^{(i)}(e^{-\varepsilon_1}, ...., e^{-\varepsilon_M})}{m_n(e^{-\varepsilon_1}, ...., e^{-\varepsilon_M})}. \tag{26}$$

Comparing Equations 22 and 26, solving for $\mu$, and using Equation 20, we find that

$$\kappa = \frac{m_n(\boldsymbol{\lambda})}{m_{n-1}^{(i)}(\boldsymbol{\lambda})} - \lambda_i \tag{27}$$

for any $i$, where $\boldsymbol{\lambda} = (\lambda_i)_{i=1}^M$. We are free to choose $i \gg n$ so that $\lambda_i$ is negligible, yielding

$$\kappa = \frac{m_n(\boldsymbol{\lambda})}{m_{n-1}(\boldsymbol{\lambda})}. \tag{28}$$

We have arrived at an *explicit* expression for $\kappa$. This expression "solves" Equation 6, the implicit equation defining $\kappa$, in the thermodynamic limit. Numerics easily confirm a close match with the implicit solution. While many works have defined similar constants implicitly, this is the first place to our knowledge that the explicit solution appears.

The meaning of $\kappa$ is elucidated by analogy to $\mu$ in the canonical ensemble. In the canonical ensemble, all orbitals are "fighting" for the supply of $n$ particles, and $\mu$ represents their "average pull" on each unit. Similarly, in KRR, we can view eigenmodes as competing for the kernel's supply of $n$ units of learnability, with $\kappa$ encoding their average pull on this supply. This analogy deepens our picture of KRR as the competition among eigenmodes for a fixed supply of learnability.

### F.2 Additional analogous quantities

In the grand canonical ensemble, the total occupation $N = \sum_i n_i$ concentrates about its mean $n$ and has fluctuations given by $\mathrm{Var}[N] = \sum_i \langle n_i \rangle (1 - \langle n_i \rangle)$ (since each site constitutes a Bernoulli variable). Comparing with equation 8, we find that

$$n/\mathcal{E}_0 \Leftrightarrow \mathrm{Var}[N]. \tag{29}$$

Smaller particle number fluctuations in the analogous physical system thus correspond to larger $\mathcal{E}_0$ and greater overfitting of noise. This provides a physical interpretation of the "overfitting as overconfidence" notion of Section 4.2.

The grand canonical partition function of our free Fermi gas is

$$Z_\mathrm{GC} = \prod_i (1 + e^{\varepsilon_i + \mu}). \tag{30}$$

The analogous quantity in KRR is

$$Z_\mathrm{KRR} = \prod_i (1 + \frac{\lambda_i}{\kappa}). \tag{31}$$

It holds that

$$\frac{\partial \ln Z_\mathrm{GC}}{\partial \mu} = \langle n \rangle, \tag{32}$$

and indeed we find that

$$-\frac{\partial \ln Z_\mathrm{KRR}}{\partial \ln \kappa} = n. \tag{33}$$

## F.3 UNIVERSAL LEARNABILITY CURVES

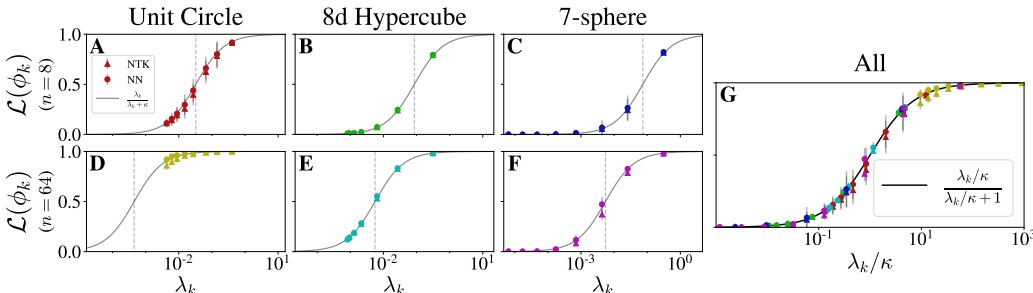

Figure 10: **Modewise learnabilities fall on universal sigmoidal curves. (A-F)** Predicted learnability curve (sigmoidal curves) and empirical learnabilities for trained networks (circles) and NTK regression (triangles) for eigenmodes $k \in \{0, ..., 7\}$ on three domains for $n = 8, 64$. Vertical dashed lines indicate $\kappa$. **(G)** All data from (A-F) with eigenvalues rescaled by $\kappa$.

Figure 10 shows that curves of learnability vs. eigenvalue collapse onto a single sigmoidal curve upon rescaling. We note that similar collapse of data from different experiments upon rescaling occurs in many important statistical physics systems including superconductors (Kardar, 2007) and turbulent flows (Goldenfeld, 2006).

It is broadly worth noting that informative constants determined by self-consistency conditions are a hallmark of statistical mechanics. It is thus sensible that such a constant would be emerge from the replica calculation of Canatar et al. (2021).

## G  PROOFS: LEARNABILITY AND ITS CONSERVATION LAW

In this appendix, we provide proofs of the formal claims of Section 3. We make use of our learning transfer matrix formalism (discussed in Appendix H.2) for these proofs as it improves clarity and compactness, but all our proofs can also be straightforwardly carried through without it. The learning transfer matrix is defined as

$$\mathbf{T}^{(\mathcal{D})} \equiv \mathbf{\Lambda}\mathbf{\Phi} \left( \mathbf{\Phi}^\top \mathbf{\Lambda} \mathbf{\Phi} + \delta \mathbf{I}_n \right)^{-1} \mathbf{\Phi}^\top, \tag{51}$$

with $\mathbf{\Phi}_{ij} = \phi_i(x_j)$ and $\mathbf{T} \equiv \mathbb{E}_{\mathcal{D}}\left[\mathbf{T}^{(\mathcal{D})}\right]$. It holds that $\hat{\mathbf{v}} = \mathbf{T}^{(\mathcal{D})}\mathbf{v}$. It is easy to see (by making $\mathbf{v}$ a one-hot vector) that $\mathcal{L}^{(\mathcal{D})}(\phi_i) = \mathbf{T}_{ii}^{(\mathcal{D})}$ and $\mathcal{L}(\phi_i) = \mathbf{T}_{ii}$.

***Property (a):*** $\mathcal{L}(\phi_i), \mathcal{L}^{(\mathcal{D})}(\phi_i) \in [0, 1]$.

Letting $\mathbf{e}_i$ be the one-hot unit vector with a one at index $i$, we observe that

$$\mathcal{L}^{(\mathcal{D})}(\phi_i) = \mathbf{e}_i^\top \mathbf{T}^{(\mathcal{D})} \mathbf{e}_i \tag{34}$$

$$= \mathbf{e}_i^\top \mathbf{\Lambda}\mathbf{\Phi} \left( \mathbf{\Phi}^\top \mathbf{\Lambda} \mathbf{\Phi} + \delta \mathbf{I}_n \right)^{-1} \mathbf{\Phi}^\top \mathbf{e}_i \tag{35}$$

$$= \lambda_i^{1/2} \mathbf{e}_i^\top \mathbf{\Phi} \left( \mathbf{\Phi}^\top \mathbf{\Lambda} \mathbf{\Phi} + \delta \mathbf{I}_n \right)^{-1} \mathbf{\Phi}^\top \mathbf{e}_i \lambda_i^{1/2} \tag{36}$$

$$= \mathbf{z}^\top \left( \mathbf{z}\mathbf{z}^\top + \mathbf{M} \right)^{-1} \mathbf{z} \qquad \in [0, 1]. \tag{37}$$

where in the third line we have used the fact that $\mathbf{e}_i$ is a unit vector and $\mathbf{\Lambda}$ is diagonal, in the fourth line we have defined $\mathbf{z} = \lambda_i^{1/2}\mathbf{\Phi}^\top \mathbf{e}_i$ and $\mathbf{M} = \mathbf{\Phi}^\top \mathbf{\Lambda} \mathbf{\Phi} + \delta \mathbf{I}_n - \mathbf{z}\mathbf{z}^\top$, and in the fifth line we have used the fact that $\mathbf{M}$ is positive semidefinite. Given this, $\mathcal{L}(\phi_i) \in [0, 1]$ by averaging.

***Property (b):*** $\mathcal{L}^{(\mathcal{D})}(f) = \mathcal{L}(f) = 0$.

*Proof.* When $n = 0$, the predicted function $\hat{f}$ is uniformly zero, and so we have $\mathcal{L}^{(\mathcal{D})}(f) = \mathcal{L}(f) = 0$.

**Remark.** As a corollary, it is easy to show that, when $M$ is finite, $n = M$, and $\delta = 0$, then $\mathcal{L}^{(\mathcal{D})}(f) = \mathcal{L}(f) = 1$.

**Property (c):** Let $\mathcal{D}_+$ be $\mathcal{D} \cup x$, where $x \in X, x \notin \mathcal{D}$ is a new data point. Then $\mathcal{L}^{(\mathcal{D}_+)}(\phi_i) \geq \mathcal{L}^{(\mathcal{D})}(\phi_i)$.

We first use the Moore-Penrose pseudoinverse, which we denote by $(\cdot)^+$, to cast $\mathbf{T}^{(\mathcal{D})}$ into the dual form

$$\mathbf{T}^{(\mathcal{D})} \equiv \boldsymbol{\Lambda}\boldsymbol{\Phi}\left(\boldsymbol{\Phi}^\top\boldsymbol{\Lambda}\boldsymbol{\Phi}\right)^{-1}\boldsymbol{\Phi}^\top = \boldsymbol{\Lambda}^{1/2}\left(\boldsymbol{\Lambda}^{1/2}\boldsymbol{\Phi}\boldsymbol{\Phi}^\top\boldsymbol{\Lambda}^{1/2}\right)\left(\boldsymbol{\Lambda}^{1/2}\boldsymbol{\Phi}\boldsymbol{\Phi}^\top\boldsymbol{\Lambda}^{1/2} + \delta\mathbf{I}_M\right)^+\boldsymbol{\Lambda}^{-1/2},$$
(38)

This follows from the property of pseudoinverses that $\mathbf{A}(\mathbf{A}^\top\mathbf{A}+\delta\mathbf{I})^+\mathbf{A}^\top = (\mathbf{A}\mathbf{A}^\top)(\mathbf{A}\mathbf{A}^\top+\delta\mathbf{I})^+$ for any matrix $\mathbf{A}$. We now augment our system with one extra data point, getting

$$\mathbf{T}^{(\mathcal{D}_+)} = \boldsymbol{\Lambda}^{1/2}\left(\boldsymbol{\Lambda}^{1/2}(\boldsymbol{\Phi}\boldsymbol{\Phi}^\top + \xi\xi^\top)\boldsymbol{\Lambda}^{1/2}\right)\left(\boldsymbol{\Lambda}^{1/2}(\boldsymbol{\Phi}\boldsymbol{\Phi}^\top + \xi\xi^\top)\boldsymbol{\Lambda}^{1/2} + \delta\mathbf{I}_M\right)^+\boldsymbol{\Lambda}^{-1/2}, \quad (39)$$

where $\xi$ is an $M$-element column vector. Equations 38 and 39 yield that

$$\mathcal{L}^{(\mathcal{D})}(\phi_i) = \mathbf{e}_i^\top\mathbf{T}^{(\mathcal{D})}\mathbf{e}_i = \mathbf{e}_i^\top\left(\boldsymbol{\Lambda}^{1/2}\boldsymbol{\Phi}\boldsymbol{\Phi}^\top\boldsymbol{\Lambda}^{1/2}\right)\left(\boldsymbol{\Lambda}^{1/2}\boldsymbol{\Phi}\boldsymbol{\Phi}^\top\boldsymbol{\Lambda}^{1/2} + \delta\mathbf{I}_M\right)^+\mathbf{e}_i, \quad (40)$$

$$\mathcal{L}^{(\mathcal{D}_+)}(\phi_i) = \mathbf{e}_i^\top\mathbf{T}^{(\mathcal{D}_+)}\mathbf{e}_i = \mathbf{e}_i^\top\left(\boldsymbol{\Lambda}^{1/2}(\boldsymbol{\Phi}\boldsymbol{\Phi}^\top + \xi\xi^\top)\boldsymbol{\Lambda}^{1/2}\right)\left(\boldsymbol{\Lambda}^{1/2}(\boldsymbol{\Phi}\boldsymbol{\Phi}^\top + \xi\xi^\top) + \delta\mathbf{I}_M\boldsymbol{\Lambda}^{1/2}\right)^+\mathbf{e}_i. \quad (41)$$

The rightmost expressions of Equations 40 and 41 both contain a factor of the form $\mathbf{B}(\mathbf{B} + \delta\mathbf{I})^+$, where $\mathbf{A}$ is a symmetric positive semidefinite matrix. An operator of this form is a projector onto the row space of $\mathbf{B}$ when $\delta = 0$ and a variant of this projector with "shrinkage" when $\delta > 0$. Comparing these equations, we find that the projectors are the same except that, in Equation 41, there is one additional dimension in the row-space and thus one new basis vector in the projector (provided $\xi$ is orthonormal to the other columns of $\boldsymbol{\Phi}$; otherwise there are zero additional dimensions).

In the case $\delta = 0$, this new basis vector cannot decrease $\mathbf{e}_i^\top\mathbf{T}^{(\mathcal{D}_+)}\mathbf{e}_i$, and thus $\mathcal{L}^{(\mathcal{D}_+)}(\phi_i) \geq \mathcal{L}^{(\mathcal{D})}(\phi_i)$ in the ridgeless case. In the case $\delta > 0$, a singular value decomposition of the projector confirms that the addition still cannot decrease $\mathbf{e}_i^\top\mathbf{T}^{(\mathcal{D}_+)}\mathbf{e}_i$. This shows the desired property. It follows as a corollary that increasing $n \to n + 1$ cannot decrease $\mathcal{L}(f)$.

**Property (d):** $\frac{\partial}{\partial\lambda_i}\mathcal{L}^{(\mathcal{D})}(\phi_i) \geq 0$, $\frac{\partial}{\partial\lambda_i}\mathcal{L}^{(\mathcal{D})}(\phi_j) \leq 0$, and $\frac{\partial}{\partial\delta}\mathcal{L}^{(\mathcal{D})}(\phi_i) \leq 0$.

*Proof.* Differentiating $\mathbf{T}_{jj}^{(\mathcal{D})}$ with respect to a particular $\lambda_i$, we find that

$$\frac{\partial}{\partial\lambda_i}\mathcal{L}^{(\mathcal{D})}(\phi_i) = \frac{\partial}{\partial\lambda_i}\mathbf{T}_{jj}^{(\mathcal{D})} = (\delta_{ij} - \lambda_j\phi_j^\top\mathbf{K}^{-1}\phi_i)\phi_i^\top\mathbf{K}^{-1}\phi_j, \quad (42)$$

where $\phi_i^\top$ is the $i$th row of $\boldsymbol{\Phi}$ and $\mathbf{K} = \boldsymbol{\Phi}^\top\boldsymbol{\Lambda}\boldsymbol{\Phi} + \delta\mathbf{I}_n$. Specializing to the case $i = j$, we note that $\phi_i^\top\mathbf{K}^{-1}\phi_i \geq 0$ because $\mathbf{K}$ is positive definite, and $\lambda_i\phi_i\mathbf{K}^{-1}\phi_i^\top \leq 1$ because $\lambda_i\phi_i\phi_i^\top$ is one of the positive semidefinite summands in $\mathbf{K} = \sum_k\lambda_k\phi_k\phi_k^\top + \delta\mathbf{I}_n$. The first clause of the property follows.

To prove the second clause, we instead specialize to the case $i \neq j$, which yields that

$$\frac{\partial}{\partial\lambda_i}\mathcal{L}^{(\mathcal{D})}(\phi_j) = \frac{\partial}{\partial\lambda_i}\mathbf{T}_{jj}^{(\mathcal{D})} = -\lambda_j\left(\phi_j^\top\mathbf{K}^{-1}\phi_i\right)^2, \quad (43)$$

which is manifestly nonpositive because $\lambda_j > 0$. The second clause follows.

Differentiating Equation 51 w.r.t. $\delta$ yields that $\frac{\partial}{\partial\delta}\mathbf{T}^{(\mathcal{D})} = -\boldsymbol{\Lambda}\boldsymbol{\Phi}\mathbf{K}^{-2}\boldsymbol{\Phi}^\top$. We then observe that

$$\frac{\partial}{\partial\delta}\mathcal{L}^{(\mathcal{D})}(\phi_i) = \mathbf{e}_i^\top\frac{\partial}{\partial\delta}\mathbf{T}^{(\mathcal{D})}\mathbf{e}_i = -\lambda_i\mathbf{e}_i^\top\boldsymbol{\Phi}\mathbf{K}^{-2}\boldsymbol{\Phi}^\top\mathbf{e}_i, \quad (44)$$

which must be nonpositive because $\lambda_i > 0$ and $\mathbf{\Phi K}^{-2}\mathbf{\Phi}^\top$ is manifestly positive definite. The desired property follows.

**Property (e):** $\mathcal{E}(f) \geq \mathcal{B}(f) \geq (1 - \mathcal{L}(f))^2$.

Noting that $\|\mathbf{v}\| = \|f\| = 1$, expected MSE is given by

$$\mathcal{E}(f) = \mathbb{E}\left[(\mathbf{v} - \hat{\mathbf{v}})^2\right] = \|\mathbf{v}\|^2 - 2\mathbf{v}^\top \mathbb{E}[\hat{\mathbf{v}}] + \mathbb{E}\left[\hat{\mathbf{v}}^2\right] = \underbrace{1 - 2\mathbf{v}^\top \mathbb{E}[\hat{\mathbf{v}}] + \|\mathbb{E}[\hat{\mathbf{v}}]\|^2}_{\text{bias } \mathcal{B}(f)} + \underbrace{\text{Var}[\|\mathbf{v}\|]}_{\text{variance } \mathcal{V}(f)} . \quad (45)$$

It is apparent that $\mathcal{E}(f) \geq \mathcal{B}(f)$. Projecting any vector onto an arbitrary unit vector can only decrease its magnitude, and so

$$\mathcal{B}(f) \geq 1 - 2\mathbf{v}^\top \mathbb{E}[\hat{\mathbf{v}}] + \mathbb{E}\left[\hat{\mathbf{v}}^\top\right] \mathbf{v}\mathbf{v}^\top \mathbb{E}[\hat{\mathbf{v}}] = \left(1 - \mathbf{v}^\top \mathbb{E}[\hat{\mathbf{v}}]\right)^2 = (1 - \mathcal{L}(f))^2. \quad \square \quad (46)$$

### G.1 Proof of Theorem 3.2 (Conservation of Learnability)

First, we note that, for any orthogonal basis $\mathcal{F}$ on $\mathcal{X}$,

$$\sum_{f \in \mathcal{F}} \mathcal{L}^{(\mathcal{D})}(f) = \sum_{\mathbf{v} \in \mathcal{V}} \frac{\mathbf{v}^\top \mathbf{T}^{(\mathcal{D})}\mathbf{v}}{\mathbf{v}^\top \mathbf{v}}, \quad (47)$$

where $\mathcal{V}$ is an orthogonal set of vectors spanning $\mathbb{R}^M$. This is equivalent to $\text{Tr}\left[\mathbf{T}^{(\mathcal{D})}\right]$. This trace is given by

$$\text{Tr}\left[\mathbf{T}^{(\mathcal{D})}\right] = \text{Tr}\left[\mathbf{\Phi}^\top \mathbf{\Lambda}\mathbf{\Phi}(\mathbf{\Phi}^\top \mathbf{\Lambda}\mathbf{\Phi} + \delta\mathbf{I}_n)^{-1}\right] = \text{Tr}\left[\mathbf{K}(\mathbf{K} + \delta\mathbf{I}_n)^{-1}\right]. \quad (48)$$

When $\delta = 0$, this trace simplifies to $\text{Tr}[\mathbf{I}_n] = n$. When $\delta > 0$, it is strictly less than $n$. This proves the theorem. $\quad \square$

**Remark.** Theorem 3.2 is a consequence of the fact that $\mathbf{T}^{(\mathcal{D})}$ is simply a projector onto an $n$-dimensional space spanned by the embeddings of the $n$ samples.

## H Derivations: the eigenlearning equations

In this appendix, we derive the eigenlearning equations for the test risk and covariance of the predicted function. Throughout this derivation, we shall prioritize clarity and interpretability over mathematical rigor, giving a derivation which can be understood without advanced mathematical tools and thereby filling a gap in the literature. For a formal derivation using random matrix theory, see Jacot et al. (2020), and for a derivation using a replica calculation of a similar level of rigor as we use here, see Canatar et al. (2021).

In the interest of clarity, we begin with a brief summary of the appendix. We begin by discussing the problem setting (Section H.1). We then define the learning transfer matrix formalism (H.2) and use it to set the ridge parameter and noise to zero (H.3). After stating our main approximation (H.4), we find the expectation of the learning transfer matrix (H.5, H.6), using our conservation law to fix $\kappa$ (H.7, H.8). Taking well-placed derivatives, we bootstrap the expectation of the learning transfer matrix to its second order statistics (H.9) and add back the ridge and noise (H.10). We conclude with various useful bounds on $\kappa$ (H.11).

### H.1 The data distribution

We shall, in this derivation, consider a slightly more general setting than discussed in the main text. In the main text, we supposed the data were drawn i.i.d. from a continuous distribution $p$ over $\mathbb{R}^d$. Here, in addition to this case, we shall also consider a setting in which the data are sampled without replacement from a *discrete* set $\mathcal{X} \subset \mathbb{R}^d$ with $|\mathcal{X}| = M$. We will refer to this as the "discrete setting" and the setting with continuous distribution as the "continuous setting." As discussed in Appendix B, the discrete setting matches several of our experiments (namely those on the discretized unit circle and on the vertices of the hypercube).

This discrete setting clearly converges to the continuous setting as $M \to \infty$: when $M \gg n^2$, the probability of sampling the same point twice is negligible (and so we can drop the "without replacement" and sample with replacement as in the continuous setting) and, by distributing $\mathcal{X}$ throughout $\mathbb{R}^d$ with point density proportional to $p$, we approach sampling from a continuous distribution. Alternatively, we could imagine $\mathcal{X}$ consists of $M$ i.i.d. samples from $p$, so that, when we later sample $n$ points from $\mathcal{X}$, they are themselves i.i.d. samples from $p$ as $M \to \infty$. It is worth noting that the data distribution in e.g. a computer vision task *is* in fact discrete because pixel values are discretized, and thus it is reasonable to work with a discrete measure.

Kernel eigenmodes in the discrete setting are defined as $M^{-1} \sum_{x' \in \mathcal{X}} K(x, x') \phi_i(x') = \lambda_i \phi_i(x)$. Note that, in this case, the number of eigenmodes is $M$, the same number as the cardinality of $\mathcal{X}$. In the continuous setting, the number of eigenmodes is infinite (though there may be only finitely many with nonzero eigenvalues), but, like Bordelon et al. (2020), we will find it useful to assume that we need only consider a finite (but very large) number $M$. This serves merely permit us to work with finite matrices (to which the standard tools of linear algebra apply) and to thereby save us the trouble of dealing with infinite and semi-infinite matrices, which require greater care. So long as $M$ is sufficiently large, we do not lose anything in discarding the exceedingly small eigenvalue tail $\lambda_{i>M}$, as is typical in the study of kernel methods and as our final results will confirm.

Our subsequent derivation will generally apply to both the discrete and continuous settings.

## H.2 THE LEARNING TRANSFER MATRIX

We begin by translating the KRR predictor into the kernel eigenbasis. Because $K(x, x') = \sum_{i=1}^{M} \lambda_i \phi_i(x) \phi_i(x')$, we can decompose the empirical kernel matrix as $\mathbf{K}_{\mathcal{D}\mathcal{D}} = \mathbf{\Phi}^\top \mathbf{\Lambda} \mathbf{\Phi}$, where $\mathbf{\Lambda} \equiv \mathrm{diag}(\lambda_1, ..., \lambda_M)$ and $\mathbf{\Phi}$ is the $M \times n$ "design matrix" given by $\mathbf{\Phi}_{ij} \equiv \phi_i(x_j)$.

The predicted function coefficients $\hat{\mathbf{v}}$ are given by

$$\hat{\mathbf{v}}_i = \langle \phi_i, \hat{f} \rangle = \lambda_i \phi_i (\mathbf{K}_{\mathcal{D}\mathcal{D}} + \delta \mathbf{I}_n)^{-1} \mathbf{\Phi}^\top \mathbf{v}, \tag{49}$$

where we have used the orthonormality of the eigenfunctions and defined $[\phi_i]_j = \phi_i(x_j)$ to be the $i$-th row of $\mathbf{\Phi}$. Stacking these coefficients into a matrix equation, we find

$$\hat{\mathbf{v}} = \mathbf{\Lambda} \mathbf{\Phi} \left( \mathbf{\Phi}^\top \mathbf{\Lambda} \mathbf{\Phi} + \delta \mathbf{I}_n \right)^{-1} \mathbf{\Phi}^\top \mathbf{v} = \mathbf{T}^{(\mathcal{D})} \mathbf{v}, \tag{50}$$

where the *learning transfer matrix*

$$\mathbf{T}^{(\mathcal{D})} \equiv \mathbf{\Lambda} \mathbf{\Phi} \left( \mathbf{\Phi}^\top \mathbf{\Lambda} \mathbf{\Phi} + \delta \mathbf{I}_n \right)^{-1} \mathbf{\Phi}^\top, \tag{51}$$

is an $M \times M$ matrix, independent of $f$, that fully describes the model's learning behavior on a training set $\mathcal{D}$[14]. The learning transfer matrix is the same as the "reconstruction operator" of Jacot et al. (2020) viewed as a finite matrix instead of a linear operator. Full understanding of the statistics of $\mathbf{T}^{(\mathcal{D})}$ will give us the statistics of $\hat{\mathbf{v}}$ and thus of $\hat{f}$, and so our main objective will be to find the mean and the covariance of $\mathbf{T}^{(\mathcal{D})}$.

## H.3 SETTING THE RIDGE AND NOISE TO ZERO

Our setting includes both a nonzero ridge parameter and nonzero noise. However, it is by this point modern folklore that many small eigenvalues together act as an effective ridge parameter equal to their sum and that power in zero-eigenvalue modes is effectively noise (see e.g. Canatar et al. (2021) for a discussion of these). Inverting these equivalences, we should expect to be able to convert $\delta$ into a small increase to many eigenvalues and convert $\epsilon^2$ to power in a zero-eigenvalue mode, and thereby permit ourselves to consider neither ridge nor noise in our derivation and add them back at the end. Here we explain our method for doing so.

The ridge parameter can be viewed as a uniform increase in all eigenvalues. We first observe that $M^{-1} \mathbf{\Phi}^\top \mathbf{\Phi} = \mathbf{I}_n$ in the discrete case. Letting $\mathbf{T}^{(\mathcal{D})}(\mathbf{\Lambda}; \delta)$ denote the learning transfer matrix with eigenvalue matrix $\mathbf{\Lambda}$ and ridge parameter $\delta$, it follows from Equation 51 and this fact that

$$\mathbf{T}^{(\mathcal{D})}(\mathbf{\Lambda}; \delta) = \mathbf{\Lambda} \left( \mathbf{\Lambda} + \frac{\delta}{M} \mathbf{I}_M \right)^{-1} \mathbf{T}^{(\mathcal{D})} \left( \mathbf{\Lambda} + \frac{\delta}{M} \mathbf{I}_M ; 0 \right). \tag{52}$$

---

[14]We take this terminology from control theory in which, for a system under study, a "transfer function" maps inputs to outputs, or driving to response.

In the continuous case, $M^{-1}\mathbf{\Phi}^\top\mathbf{\Phi} \to \mathbf{I}_n$ as $M \to \infty$ (since the columns of $\mathbf{\Phi}$ are uncorrelated), and since $M$ is very large in the continuous setting, we are again free to use Equation 52.

As for the noise, we simply set $\epsilon^2 = 0$ for now and, once we have our final equations, add power $\epsilon^2$ to a hypothetical zero-eigenvalue mode.

### H.4   ASSUMPTION: THE UNIVERSALITY OF $\mathbf{\Phi}$

We wish to take averages over $\mathbf{\Phi}$ in finding the statistics of $\mathbf{T}^{(\mathcal{D})}$. The distribution of $\mathbf{\Phi}$ is in fact highly structured (reflecting the eigenstructure of the kernel), and we know only that $\mathbb{E}[\mathbf{\Phi}_{ij}\mathbf{\Phi}_{ij'}] = \delta_{jj'}$. We neglect this structure, making the "universality" assumption that we may take $\mathbf{\Phi}$ to be sampled from a simple Gaussian measure without substantially changing the statistics of $\mathbf{T}^{(\mathcal{D})}$. We henceforth assume $\mathbf{\Phi}_{ij} \overset{\text{iid}}{\sim} \mathcal{N}(0, 1)$.

This universality assumption is also made in comparable works (implicitly by Bordelon et al. (2020); Canatar et al. (2021) and explicitly by Jacot et al. (2020)). See Appendix A for further references to relevant literature. The validity of this approximation is ultimately justified by the close match of our theory with experiment.

### H.5   VANISHING OFF-DIAGONALS OF $\mathbb{E}\left[\mathbf{T}^{(\mathcal{D})}\right]$

We next observe that

$$\mathbb{E}_{\mathbf{\Phi}}\left[\mathbf{\Lambda}\mathbf{\Phi}\left(\mathbf{\Phi}^\top\mathbf{U}^\top\mathbf{\Lambda}\mathbf{U}\mathbf{\Phi}\right)^{-1}\mathbf{\Phi}^\top\right] = \mathbb{E}_{\mathbf{\Phi}}\left[\mathbf{\Lambda}\mathbf{U}^\top\mathbf{\Phi}\left(\mathbf{\Phi}^\top\mathbf{\Lambda}\mathbf{\Phi}\right)^{-1}\mathbf{\Phi}^\top\mathbf{U}\right], \tag{53}$$

where $\mathbf{U}$ is any orthogonal $M \times M$ matrix. Defining $\mathbf{U}^{(m)}$ as the matrix such that $\mathbf{U}^{(m)}_{ab} \equiv \delta_{ab}(1 - 2\delta_{am})$, noting that $\mathbf{U}^{(m)}\mathbf{\Lambda}\mathbf{U}^{(m)} = \mathbf{\Lambda}$, and plugging $\mathbf{U}^{(m)}$ in as $\mathbf{U}$ in Equation 53, we find that

$$\mathbb{E}\left[\mathbf{T}^{(\mathcal{D})}_{ab}\right] = \left(\left(\mathbf{U}^{(m)}\right)^\top \mathbb{E}\left[\mathbf{T}^{(\mathcal{D})}\right]\mathbf{U}^{(m)}\right)_{ab} = (-1)^{\delta_{am}+\delta_{bm}}\mathbb{E}\left[\mathbf{T}^{(\mathcal{D})}_{ab}\right]. \tag{54}$$

By choosing $m = a$, we conclude that $\mathbb{E}\left[\mathbf{T}^{(\mathcal{D})}_{ab}\right] = 0$ if $a \neq b$.

### H.6   FIXING THE FORM OF $\mathbb{E}\left[\mathbf{T}^{(\mathcal{D})}_{ii}\right]$

#### H.6.1   ISOLATING THE DESIRED ELEMENT

We now isolate a particular diagonal element of the mean learning transfer matrix. To do so, we write $\mathbb{E}\left[\mathbf{T}^{(\mathcal{D})}_{ii}\right]$ in terms of $\lambda_i$ (the $i$th eigenvalue), $\mathbf{\Lambda}_{(i)}$ ($\mathbf{\Lambda}$ with its $i$th row and column removed), $\phi_i^\top$ (the $i$th row of $\mathbf{\Phi}$), and $\mathbf{\Phi}_{(i)}$ ($\mathbf{\Phi}$ with its $i$th row removed).

Using the Sherman-Morrison matrix inversion formula, we find that

$$\left(\mathbf{\Phi}^\top\mathbf{\Lambda}\mathbf{\Phi}\right)^{-1} = \left(\mathbf{\Phi}_{(i)}^\top\mathbf{\Lambda}_{(i)}\mathbf{\Phi}_{(i)} + \lambda_i\phi_i\phi_i^\top\right)^{-1}$$
$$= \left(\mathbf{\Phi}_{(i)}^\top\mathbf{\Lambda}_{(i)}\mathbf{\Phi}_{(i)}\right)^{-1} - \frac{\lambda_i\left(\mathbf{\Phi}_{(i)}^\top\mathbf{\Lambda}_{(i)}\mathbf{\Phi}_{(i)}\right)^{-1}\phi_i\phi_i^\top\left(\mathbf{\Phi}_{(i)}^\top\mathbf{\Lambda}_{(i)}\mathbf{\Phi}_{(i)}\right)^{-1}}{1 + \lambda_i\phi_i^\top\left(\mathbf{\Phi}_{(i)}^\top\mathbf{\Lambda}_{(i)}\mathbf{\Phi}_{(i)}\right)^{-1}\phi_i}. \tag{55}$$

Inserting this into the expectation of $\mathbf{T}^{(\mathcal{D})}$, we find that

$$
\begin{aligned}
\mathbb{E}\left[\mathbf{T}_{ii}^{(\mathcal{D})}\right] &= \mathbb{E}_{\mathbf{\Phi}_{(i)},\phi_i}\left[\lambda_i\phi_i^\top\left(\mathbf{\Phi}^\top\mathbf{\Lambda}\mathbf{\Phi}\right)^{-1}\phi_i\right] \\
&= \mathbb{E}_{\mathbf{\Phi}_{(i)},\phi_i}\left[\lambda_i\phi_i^\top\left(\mathbf{\Phi}_{(i)}^\top\mathbf{\Lambda}_{(i)}\mathbf{\Phi}_{(i)}\right)^{-1}\phi_i - \frac{\lambda_i^2\left[\phi_i^\top\left(\mathbf{\Phi}_{(i)}^\top\mathbf{\Lambda}_{(i)}\mathbf{\Phi}_{(i)}\right)^{-1}\phi_i\right]^2}{1+\lambda_i\phi_i^\top\left(\mathbf{\Phi}_{(i)}^\top\mathbf{\Lambda}_{(i)}\mathbf{\Phi}_{(i)}\right)^{-1}\phi_i}\right] \\
&= \mathbb{E}_{\mathbf{\Phi}_{(i)},\phi_i}\left[\frac{\lambda_i}{\lambda_i+\left[\phi_i^\top\left(\mathbf{\Phi}_{(i)}^\top\mathbf{\Lambda}_{(i)}\mathbf{\Phi}_{(i)}\right)^{-1}\phi_i\right]^{-1}}\right] \\
&= \mathbb{E}_{\mathbf{\Phi}_{(i)},\phi_i}\left[\frac{\lambda_i}{\lambda_i+\kappa^{(\mathbf{\Phi}_{(i)},\phi_i)}}\right],
\end{aligned}
\tag{56}
$$

where $\kappa^{(\mathbf{\Phi}_{(i)},\phi_i)} \equiv \kappa_i^{(\mathbf{\Phi})} \equiv \left[\phi_i^\top\left(\mathbf{\Phi}_{(i)}^\top\mathbf{\Lambda}_{(i)}\mathbf{\Phi}_{(i)}\right)^{-1}\phi_i\right]^{-1}$ is a nonnegative scalar.

## H.7 Concentration and mode-independence of $\kappa_i^{(\mathbf{\Phi})}$

Important quantities in statistical mechanics systems are typically self-averaging (i.e. concentrating about their expectation) in the thermodynamic limit. Self-averaging quantities are the focus of random matrix theory, and generalization metrics in machine learning also tend to be self-averaging under most circumstances (e.g. resampling the data and rerunning a training procedure will typically yield a similar generalization error). Here we argue that $\kappa_i^{(\mathbf{\Phi})}$ is self-averaging in the thermodynamic limit and can be replaced by its expectation.

This could be shown rigorously by means of random matrix theory. Here we opt to simply observe that, if $\kappa_i^{(\mathbf{\Phi})}$ were *not* self-averaging, then for modes $i$ such that $\lambda_i \sim \kappa_i^{(\mathbf{\Phi})}$, $\mathbf{T}_{ii}^{(\mathcal{D})}$ and thus $\mathcal{L}^{(\mathcal{D})}(\phi_i)$ would not be self-averaging either. However, because $\mathcal{L}^{(\mathcal{D})}(\phi_i)$ is a generalization metric like MSE, we should in general expect that it is self-averaging at large $n$. Our experimental results (Figure 2) confirm that fluctuations in $\mathcal{L}^{(\mathcal{D})}(\phi_i)$ are indeed small in practice, especially at large $n$. We thus replace $\kappa_i^{(\mathbf{\Phi})}$ with its expectation $\kappa_i \equiv \mathbb{E}_{\mathbf{\Phi}}\left[\kappa_i^{(\mathbf{\Phi})}\right]$.

We next argue that $\kappa_i$ is approximately independent of $i$, so we can replace it with a mode-independent constant $\kappa$. This, too, could be argued rigorously by means of random matrix theory. We opt instead for an eigenmode-removal argument inspired by the cavity method of statistical physics (Del Ferraro et al., 2014). Observe that, in the thermodynamic limit, the addition or removal of a single eigenmode should have a negligible effect on any observable and thus on $\kappa_i$. We shall here show that, by inserting one eigenmode and removing another, we can transform $\kappa_i$ into $\kappa_j$ for any $i$ and $j$, implying that $\kappa_i \approx \kappa_j$.

Assume that the addition or removal of a single eigenmode negligibly affects $\kappa_i$. Concretely, assume that $\kappa_i \approx \kappa_i^+$, where $\kappa_i^+$ is $\kappa_i$ computed with the addition of one extra eigenmode of arbitrary eigenvalue. We choose the additional eigenmode to have eigenvalue $\lambda_i$, and we insert it at index $i$, effectively reinserting the missing mode $i$ into $\mathbf{\Phi}_{(i)}$ and $\mathbf{\Lambda}_{(i)}$.

To clarify the random variables in play, we shall adopt a more explicit notation, writing out $\mathbf{\Phi}_{(i)}$ in terms of its row vectors as $\mathbf{\Phi}_{(i)} = [\phi_1, ..., \phi_{i-1}, \phi_{i+1}, ..., \phi_M]^\top$. Using this notation, we find upon adding the new eigenmode that

$$\kappa_i \equiv \mathbb{E}_{\mathbf{\Phi}_{(i)}, \phi_i} \left[ \left[ \phi_i^\top \left( \mathbf{\Phi}_{(i)}^\top \mathbf{\Lambda}_{(i)} \mathbf{\Phi}_{(i)} \right)^{-1} \phi_i \right]^{-1} \right] \tag{57}$$

$$\equiv \mathbb{E}_{\{\phi_k\}_{k=1}^M} \left[ \left[ \phi_i^\top \left( [\phi_1, ..., \phi_{i-1}, \phi_{i+1}, ..., \phi_M] \mathbf{\Lambda}_{(i)} [\phi_1, ..., \phi_{i-1}, \phi_{i+1}, ..., \phi_M]^\top \right)^{-1} \phi_i \right]^{-1} \right] \tag{58}$$

$$\approx \kappa_i^+ \equiv \mathbb{E}_{\{\phi_k\}_{k=1}^M, \tilde{\phi}_i} \left[ \left[ \phi_i^\top \left( [\phi_1, ..., \phi_{i-1}, \tilde{\phi}_i, \phi_{i+1}, ..., \phi_M] \mathbf{\Lambda} [\phi_1, ..., \phi_{i-1}, \tilde{\phi}_i, \phi_{i+1}, ..., \phi_M]^\top \right)^{-1} \phi_i \right]^{-1} \right], \tag{59}$$

where $\mathbf{\Lambda}$ is the original eigenvalue matrix and $\tilde{\phi}_i^\top$ is the design matrix row corresponding to the new mode.

We can also perform the same manipulation with $\kappa_j$ ($j \neq i$), this time adding an additional eigenvalue $\lambda_j$ at index $j$, yielding that

$$\kappa_j \equiv \mathbb{E}_{\mathbf{\Phi}_{(j)}, \phi_j} \left[ \left[ \phi_j^\top \left( \mathbf{\Phi}_{(j)}^\top \mathbf{\Lambda}_{(j)} \mathbf{\Phi}_{(j)} \right)^{-1} \phi_j \right]^{-1} \right] \tag{60}$$

$$\equiv \mathbb{E}_{\{\phi_k\}_{k=1}^M} \left[ \left[ \phi_j^\top \left( [\phi_1, ..., \phi_{j-1}, \phi_{j+1}, ..., \phi_M] \mathbf{\Lambda}_{(j)} [\phi_1, ..., \phi_{j-1}, \phi_{j+1}, ..., \phi_M]^\top \right)^{-1} \phi_j \right]^{-1} \right] \tag{61}$$

$$\approx \kappa_j^+ \equiv \mathbb{E}_{\{\phi_k\}_{k=1}^M, \tilde{\phi}_j} \left[ \left[ \phi_j^\top \left( [\phi_1, ..., \phi_{j-1}, \tilde{\phi}_j, \phi_{j+1}, ..., \phi_M] \mathbf{\Lambda} [\phi_1, ..., \phi_{j-1}, \tilde{\phi}_j, \phi_{j+1}, ..., \phi_M]^\top \right)^{-1} \phi_j \right]^{-1} \right]. \tag{62}$$

We now compare Equations 59 and 62. Each is an expectation over $M+1$ vectors from the isotropic measure . T he statistics of these $M+1$ vectors are symmetric under exchange, so we are free to relabel them. Equation 59 is identical to Equation 62 upon relabeling $\phi_i \to \phi_j$, $\tilde{\phi}_i \to \phi_i$, and $\phi_j \to \tilde{\phi}_j$, so they are equivalent, and $\kappa_i^+ = \kappa_j^+$. This in turn implies that $\kappa_i \approx \kappa_j$. In light of this, we now replace all $\kappa_i$ with a mode-independent (but as-yet-unknown) constant $\kappa$ and conclude that

$$\boxed{\mathbb{E}\left[ \mathbf{T}_{ij}^{(\mathcal{D})} \right] = \delta_{ij} \frac{\lambda_i}{\lambda_i + \kappa}.} \tag{63}$$

In summary, we have argued that $\kappa_i$ is not significantly changed by the addition or removal of single eigenmodes, and two such changes permit us to transform $\kappa_i$ into $\kappa_j$, so they are therefore approximately equal.

Our argument here is similar to the cavity method of statistical physics (Del Ferraro et al., 2014), which essentially compares the behavior of a weakly-interacting system with and without a single element. The cavity method is often used as a simpler and more intuitive alternative to the replica method, a role it reprises here (contrast our approach with the replica approach of Canatar et al. (2021)).

## H.8  DETERMINING $\kappa$

We can determine the value of $\kappa$ by observing that, using the ridgeless case of Theorem 3.2,

$$\sum_i \mathbb{E}\left[ \mathbf{T}_{ii}^{(\mathcal{D})} \right] = \sum_i \frac{\lambda_i}{\lambda_i + \kappa} = n. \tag{64}$$

This is a much more straightforward method of fixing this constant than used in comparable works. The ability to use the ridgeless version of Theorem 3.2 is the main motivation for setting $\delta = 0$ at the start of the derivation.

## H.9 DIFFERENTIATING W.R.T. $\mathbf{\Lambda}$ TO OBTAIN THE COVARIANCE OF $\mathbf{T}^{(\mathcal{D})}$

Here we obtain expressions for the covariance of $\mathbf{T}^{(\mathcal{D})}$ and thereby the covariance of $\hat{\mathbf{v}}$. Remarkably, we shall need no further approximations beyond those already made in approximating $\mathbb{E}\left[\mathbf{T}^{(\mathcal{D})}\right]$, which lends credence to our thesis that understanding modewise learnabilities is sufficient for understanding more interesting statistics of $\hat{f}$.

We begin with a calculation that will later be of use: differentiating both sides of the constraint on $\kappa$ with respect to a particular eigenvalue, we find that

$$\frac{\partial}{\partial \lambda_i} \sum_{j=1}^{M} \frac{\lambda_j}{\lambda_j + \kappa} = \sum_{j=1}^{M} \frac{-\lambda_j}{(\lambda_j + \kappa)^2} \frac{\partial \kappa}{\partial \lambda_i} + \frac{\kappa}{(\lambda_i + \kappa)^2} = 0, \tag{65}$$

yielding that

$$\frac{\partial \kappa}{\partial \lambda_i} = \frac{\kappa^2}{q(\lambda_i + \kappa)^2} \quad \text{where} \quad q \equiv \sum_{j=1}^{M} \frac{\kappa \lambda_j}{(\lambda_j + \kappa)^2}. \tag{66}$$

We now factor $\mathbf{T}^{(\mathcal{D})}$ into two matrices as

$$\mathbf{T}^{(\mathcal{D})} = \mathbf{\Lambda}\mathbf{Z}, \quad \text{where} \quad \mathbf{Z} \equiv \mathbf{\Phi}\left(\mathbf{\Phi}^\top \mathbf{\Lambda} \mathbf{\Phi}\right)^{-1} \mathbf{\Phi}^\top. \tag{67}$$

Unlike $\mathbf{T}^{(\mathcal{D})}$, the matrix $\mathbf{Z}$ has the advantage of being symmetric and containing only one factor of $\mathbf{\Lambda}$. We will find the second-order statistics of $\mathbf{Z}$, which will trivially give these statistics for $\mathbf{T}^{(\mathcal{D})}$. From Equation 63, we find that the expectation of $\mathbf{Z}$ is

$$\mathbb{E}[\mathbf{Z}] = (\mathbf{\Lambda} + \kappa \mathbf{I}_M)^{-1}. \tag{68}$$

We also define a modified $\mathbf{Z}$-matrix $\mathbf{Z}^{(\mathbf{U})} \equiv \mathbf{\Phi}\left(\mathbf{\Phi}^\top \mathbf{U}^\top \mathbf{\Lambda} \mathbf{U} \mathbf{\Phi}\right)^{-1} \mathbf{\Phi}^\top$, where $\mathbf{U}$ is an orthogonal $M \times M$ matrix. Because the measure over which $\mathbf{\Phi}$ is averaged is rotation-invariant, we can equivalently average over $\tilde{\mathbf{\Phi}} \equiv \mathbf{U}\mathbf{\Phi}$ with the same measure, giving

$$\mathbb{E}_{\mathbf{\Phi}}\left[\mathbf{Z}^{(\mathbf{U})}\right] = \mathbb{E}_{\tilde{\mathbf{\Phi}}}\left[\mathbf{U}^\top \tilde{\mathbf{\Phi}} \left(\tilde{\mathbf{\Phi}}^\top \mathbf{\Lambda} \tilde{\mathbf{\Phi}}\right)^{-1} \tilde{\mathbf{\Phi}}^\top \mathbf{U}\right] = \mathbb{E}_{\mathbf{\Phi}}\left[\mathbf{U}^\top \mathbf{Z} \mathbf{U}\right] = \mathbf{U}^\top (\mathbf{\Lambda} + \kappa \mathbf{I}_M)^{-1} \mathbf{U}. \tag{69}$$

It is similarly the case that

$$\mathbb{E}_{\mathbf{\Phi}}\left[(\mathbf{Z}^{(\mathbf{U})})_{ij}(\mathbf{Z}^{(\mathbf{U})})_{k\ell}\right] = \mathbb{E}_{\mathbf{\Phi}}\left[\left(\mathbf{U}^\top \mathbf{Z} \mathbf{U}\right)_{ij} \left(\mathbf{U}^\top \mathbf{Z} \mathbf{U}\right)_{k\ell}\right]. \tag{70}$$

Our aim will be to calculate expectations of the form $\mathbb{E}_{\mathbf{\Phi}}[\mathbf{Z}_{ij}\mathbf{Z}_{k\ell}]$. With a clever choice of $\mathbf{U}$, a symmetry argument quickly shows that most choices of the four indices make this expression zero. We define $\mathbf{U}_{ab}^{(m)} \equiv \delta_{ab}(1 - 2\delta_{am})$ and observe that, because $\mathbf{\Lambda}$ is diagonal, $(\mathbf{U}^{(m)})^\top \mathbf{\Lambda} \mathbf{U}^{(m)} = \mathbf{\Lambda}$ and thus $\mathbf{Z}^{(\mathbf{U}^{(m)})} = \mathbf{Z}$. Equation 70 then yields that

$$\mathbb{E}_{\mathbf{\Phi}}[\mathbf{Z}_{ij}\mathbf{Z}_{k\ell}] = (-1)^{\delta_{im}+\delta_{jm}+\delta_{km}+\delta_{\ell m}} \mathbb{E}_{\mathbf{\Phi}}[\mathbf{Z}_{ij}\mathbf{Z}_{k\ell}], \tag{71}$$

from which it follows that $\mathbb{E}_{\mathbf{\Phi}}[\mathbf{Z}_{ij}\mathbf{Z}_{k\ell}] = 0$ if any index is repeated an odd number of times. In light of the fact that $\mathbf{Z}_{ij} = \mathbf{Z}_{ji}$, there are only three distinct nontrivial cases to consider:

1. $\mathbb{E}_{\mathbf{\Phi}}[\mathbf{Z}_{ii}\mathbf{Z}_{ii}]$,
2. $\mathbb{E}_{\mathbf{\Phi}}[\mathbf{Z}_{ij}\mathbf{Z}_{ij}]$ with $i \neq j$, and
3. $\mathbb{E}_{\mathbf{\Phi}}[\mathbf{Z}_{ii}\mathbf{Z}_{jj}]$ with $i \neq j$.

We note that we are not using the Einstein convention of summation over repeated indices.

**Cases 1 and 2.** We now consider differentiating $\mathbf{Z}$ with respect to a particular element of the matrix $\boldsymbol{\Lambda}$. This yields

$$\frac{\partial \mathbf{Z}_{i\ell}}{\partial \boldsymbol{\Lambda}_{jk}} = -\boldsymbol{\phi}_i^\top \left(\boldsymbol{\Phi}^\top \boldsymbol{\Lambda} \boldsymbol{\Phi}\right)^{-1} \boldsymbol{\phi}_j \boldsymbol{\phi}_k^\top \left(\boldsymbol{\Phi}^\top \boldsymbol{\Lambda} \boldsymbol{\Phi}\right)^{-1} \boldsymbol{\phi}_\ell = -\mathbf{Z}_{ij}\mathbf{Z}_{k\ell}, \tag{72}$$

where as before $\boldsymbol{\phi}_i$ is the $i$-th row of $\Phi$. This gives us the useful expression that

$$\mathbb{E}[\mathbf{Z}_{ij}\mathbf{Z}_{k\ell}] = -\frac{\partial}{\partial \boldsymbol{\Lambda}_{jk}} \mathbb{E}[\mathbf{Z}_{i\ell}]. \tag{73}$$

We now set $\ell = i, k = j$ and evaluate this expression using Equation 68, concluding that

$$\mathbb{E}[\mathbf{Z}_{ij}\mathbf{Z}_{ij}] = \mathbb{E}[\mathbf{Z}_{ij}\mathbf{Z}_{ji}] = -\frac{\partial}{\partial \lambda_j}\left(\frac{1}{\lambda_i + \kappa}\right) = \frac{1}{(\lambda_i + \kappa)^2}\left(\delta_{ij} + \frac{\partial \kappa}{\partial \lambda_j}\right) \tag{74}$$

and thus

$$\mathrm{Cov}[\mathbf{Z}_{ij}, \mathbf{Z}_{ij}] = \mathrm{Cov}[\mathbf{Z}_{ij}, \mathbf{Z}_{ji}] = \frac{1}{(\lambda_i + \kappa)^2}\frac{\partial \kappa}{\partial \lambda_j} = \frac{\kappa^2}{q(\lambda_i + \kappa)^2(\lambda_j + \kappa)^2}. \tag{75}$$

We did not require that $i \neq j$, and so Equation 74 holds for Case 1 as well as Case 2.

**Case 3.** We now aim to calculate $\mathbb{E}[\mathbf{Z}_{ii}\mathbf{Z}_{jj}]$ for $i \neq j$. We might hope to use Equation 73 in calculating $\mathbb{E}[\mathbf{Z}_{ii}\mathbf{Z}_{jj}]$, but this approach is stymied by the fact that we would need to take a derivative with respect to $\boldsymbol{\Lambda}_{ij}$, but we only have an approximation for $\mathbf{Z}$ for diagonal $\boldsymbol{\Lambda}$. We can circumvent this by means of $\mathbf{Z}^{(\mathbf{U})}$. From the definition of $\mathbf{Z}^{(\mathbf{U})}$, we find that

$$\left(\frac{\partial}{\partial \mathbf{U}_{ij}} - \frac{\partial}{\partial \mathbf{U}_{ji}}\right)\mathbf{Z}^{(\mathbf{U})}\bigg|_{\mathbf{U}=\mathbf{I}_M}$$

$$= -\boldsymbol{\phi}_i^\top \left(\boldsymbol{\Phi}^\top \boldsymbol{\Lambda} \boldsymbol{\Phi}\right)^{-1}\left[\boldsymbol{\phi}_j \lambda_i \boldsymbol{\phi}_i^\top - \boldsymbol{\phi}_i \lambda_j \boldsymbol{\phi}_j^\top + \boldsymbol{\phi}_i \lambda_i \boldsymbol{\phi}_j^\top - \boldsymbol{\phi}_j \lambda_j \boldsymbol{\phi}_i^\top\right]\left(\boldsymbol{\Phi}^\top \boldsymbol{\Lambda} \boldsymbol{\Phi}\right)^{-1}\boldsymbol{\phi}_j$$

$$= (\lambda_j - \lambda_i)\left(\mathbf{Z}_{ij}^2 + \mathbf{Z}_{ii}\mathbf{Z}_{jj}\right). \tag{76}$$

Differentiating with respect to both $\mathbf{U}_{ij}$ and $\mathbf{U}_{ji}$ with opposite signs ensures that the derivative is taken within the manifold of orthogonal matrices. Now, using Equation 69, we find that

$$\left(\frac{\partial}{\partial \mathbf{U}_{ij}} - \frac{\partial}{\partial \mathbf{U}_{ji}}\right)\mathbb{E}\left[\mathbf{Z}^{(\mathbf{U})}\right]\bigg|_{\mathbf{U}=\mathbf{I}_M} = \left(\frac{\partial}{\partial \mathbf{U}_{ij}} - \frac{\partial}{\partial \mathbf{U}_{ji}}\right)\mathbf{U}^\top(\boldsymbol{\Lambda} + \kappa\mathbf{I}_M)^{-1}\mathbf{U}\bigg|_{\mathbf{U}=\mathbf{I}_M}$$

$$= \frac{1}{\lambda_i + \kappa} - \frac{1}{\lambda_j + \kappa}. \tag{77}$$

Taking the expectation of Equation 76, plugging in Equation 74 for $\mathbb{E}\left[\mathbf{Z}_{ij}^2\right]$, comparing to 77, and performing some algebra, we conclude that

$$\mathbb{E}[\mathbf{Z}_{ii}\mathbf{Z}_{jj}] = \frac{1}{(\lambda_i + \kappa)(\lambda_j + \kappa)} - \frac{\kappa^2}{q(\lambda_i + \kappa)^2(\lambda_j + \kappa)^2} \tag{78}$$

and thus that $\mathbf{Z}_{ii}, \mathbf{Z}_{jj}$ are anticorrelated with covariance

$$\text{Cov}[\mathbf{Z}_{ii}, \mathbf{Z}_{jj}] = -\frac{\kappa^2}{q(\lambda_i + \kappa)^2(\lambda_j + \kappa)^2}. \tag{79}$$

Cases 1-3 can be summarized as

$$\text{Cov}[\mathbf{Z}_{ij}, \mathbf{Z}_{k\ell}] = \frac{\kappa^2 \left(\delta_{ik}\delta_{j\ell} + \delta_{i\ell}\delta_{jk} - \delta_{ij}\delta_{k\ell}\right)}{q(\lambda_i + \kappa)(\lambda_j + \kappa)(\lambda_k + \kappa)(\lambda_\ell + \kappa)}. \tag{80}$$

Using the fact that $\mathbf{T}_{ij}^{(\mathcal{D})} = \lambda_i \mathbf{Z}_{ij}$, defining $\mathcal{L}_i \equiv \lambda_i(\lambda_i + \kappa)^{-1}$, and noting that $q = (\sum_i \mathcal{L}_i(1 - \mathcal{L}_i))^{-1}$, we find that

$$\boxed{\text{Cov}\left[\mathbf{T}_{ij}^{(\mathcal{D})}, \mathbf{T}_{k\ell}^{(\mathcal{D})}\right] = \frac{\mathcal{L}_i(1 - \mathcal{L}_j)\mathcal{L}_k(1 - \mathcal{L}_\ell)}{n - \sum_{m=1}^{M} \mathcal{L}_m^2}(\delta_{ik}\delta_{j\ell} + \delta_{i\ell}\delta_{jk} - \delta_{ij}\delta_{k\ell}).} \tag{81}$$

Noting that $\mathcal{E}(f) = \mathbb{E}\left[|\mathbf{v} - \hat{\mathbf{v}}|^2\right]$, recalling that $\hat{\mathbf{v}} = \mathbf{T}^{(\mathcal{D})}\mathbf{v}$, and using Equation 81 to evaluate a sum over eigenmodes, we find that expected MSE is given by

$$\mathcal{E}(f) = \frac{n}{n - \sum_m \mathcal{L}_m^2} \sum_i (1 - \mathcal{L}_i)^2 \mathbf{v}_i^2. \tag{82}$$

Taking a sum over indices of $\mathbf{v}$, we find that the covariance of the predicted function can be written simply in terms of MSE as

$$\text{Cov}[\hat{\mathbf{v}}_i, \hat{\mathbf{v}}_j] = \frac{\mathcal{L}_i^2 \mathcal{E}(f)}{n} \delta_{ij}. \tag{83}$$

### H.10   ADDING BACK THE RIDGE AND NOISE

We have thus far assumed $\delta = 0$. We can now add the ridge parameter back using Equation 52. To add a ridge parameter $\delta$, we need merely replace $\lambda_i \to \lambda_i + \frac{\delta}{M}$ and then change $\mathbf{T}_{ij}^{(\mathcal{D})} \to \lambda_i(\lambda_i + \frac{\delta}{M})^{-1}\mathbf{T}_{ij}^{(\mathcal{D})}$. This yields that

$$\mathbb{E}\left[\mathbf{T}_{ii}^{(\mathcal{D})}\right] = \frac{\delta_{ij}\lambda_i}{\lambda_i + \frac{\delta}{M} + \kappa}, \tag{84}$$

$$\text{Cov}\left[\mathbf{T}_{ij}^{(\mathcal{D})}, \mathbf{T}_{k\ell}^{(\mathcal{D})}\right] = \frac{\kappa \left(\delta_{ik}\delta_{j\ell} + \delta_{i\ell}\delta_{jk} - \delta_{ij}\delta_{k\ell}\right)}{q(\lambda_i + \frac{\delta}{M} + \kappa)(\lambda_j + \frac{\delta}{M} + \kappa)(\lambda_k + \frac{\delta}{M} + \kappa)(\lambda_\ell + \frac{\delta}{M} + \kappa)}, \tag{85}$$

where $\kappa \geq 0$ satisfies $\displaystyle\sum_i \frac{\lambda_i + \frac{\delta}{M}}{\lambda_i + \frac{\delta}{M} + \kappa} = n$ and $q \equiv \displaystyle\sum_{j=1}^{M} \frac{\lambda_j + \frac{\delta}{M}}{(\lambda_j + \frac{\delta}{M} + \kappa)^2}. \tag{86}$

Taking either $\delta = 0$ or $M \to \infty$, we find that

$$\boxed{n = \sum_i \frac{\lambda_i}{\lambda_i + \kappa} + \frac{\delta}{\kappa}} \tag{87}$$

and the mean and covariance of $\mathbf{T}^{(\mathcal{D})}$ are again given by Equations 63 and 81.

To summarize this simplification: in the continuous setting ($M \to \infty$), we recover the results of prior work. In the discrete setting with zero ridge, we find that these expressions apply unmodified. In the discrete setting with positive ridge, we find that these expressions contain corrections with perturbative parameter $\frac{\delta}{M}$. In the main text, we report the expressions with $\frac{\delta}{M} = 0$, and our experiments obey this condition.

Modifying $\mathbf{v}$ so as to place power $\epsilon^2$ into an eigenmode with zero eigenvalue, Equation 82 for MSE becomes

$$\mathcal{E}(f) = \mathcal{E}_0 \left( \sum_i (1 - \mathcal{L}_i)^2 \mathbf{v}_i^2 + \epsilon^2 \right). \tag{88}$$

### H.11 PROPERTIES OF $\kappa$

In experimental settings, $\kappa$ is in general easy to find numerically, but for theoretical study, we anticipate it being useful to have some analytical bounds on $\kappa$ in order to, for example, prove that certain eigenmodes are or are not asymptotically learned for particular spectra. To that end, the following lemma gives some properties of $\kappa$.

**Lemma H.1.** *For $\kappa \geq 0$ solving $\sum_{i=1}^M \frac{\lambda_i}{\lambda_i + \kappa} + \frac{\delta}{\kappa} = n$, with positive eigenvalues $\{\lambda_i\}_{i=1}^M$ ordered from greatest to least, the following properties hold:*

*(a) $\kappa = \infty$ when $n = 0$, and $\kappa = 0$ when $n \to M$ and $\delta = 0$.*

*(b) $\kappa$ is strictly decreasing with $n$.*

*(c) $\kappa \leq \frac{1}{n-\ell} \left( \delta + \sum_{i=\ell+1}^M \lambda_i \right)$ for all $\ell \in \{0, ..., n-1\}$.*

*(d) $\kappa \geq \lambda_\ell \left( \frac{\ell}{n} - 1 \right)$ for all $\ell \in \{n, ..., M\}$.*

*Proof of property (a)*: Because $\sum_{i=1}^M \frac{\lambda_i}{\lambda_i + \kappa} + \frac{\delta}{\kappa}$ is strictly decreasing with $\kappa$ for $\kappa \geq 0$, there can only be one solution for a given $n$. The first statement follows by inspection, and the second follows by inspection and our assumption that all eigenvalues are strictly positive.

*Proof of property (b)*: Differentiating the constraint on $\kappa$ with respect to $n$ yields

$$\left[ \sum_{i=1}^M \frac{-\lambda_i}{(\lambda_i + \kappa)^2} + \frac{-\delta}{\kappa^2} \right] \frac{d\kappa}{dn} = 1, \quad \text{which implies that} \quad \frac{d\kappa}{dn} = -\left[ \sum_{i=1}^M \frac{\lambda_i}{(\lambda_i + \kappa)^2} + \frac{\delta}{\kappa^2} \right]^{-1} < 0. \tag{89}$$

*Proof of property (c)*: We observe that $n = \sum_{i=1}^M \frac{\lambda_i}{\lambda_i + \kappa} + \frac{\delta}{\kappa} \leq \ell + \sum_{i=\ell+1}^M \frac{\lambda_i}{\kappa} + \frac{\delta}{\kappa}$. The desired property follows.

*Proof of property (d)*: We set $\delta = 0$ and consider replacing $\lambda_i$ with $\lambda_\ell$ if $i \leq \ell$ and $0$ if $i > \ell$. Noting that this does not increase any term in the sum, we find that $n = \sum_{i=1}^M \frac{\lambda_i}{\lambda_i + \kappa} \geq \sum_{i=1}^\ell \frac{\lambda_\ell}{\lambda_\ell + \kappa} = \frac{\ell \lambda_\ell}{\lambda_\ell + \kappa}$. The desired property in the ridgeless case follows. A positive ridge parameter only increases $\kappa$, so the property holds in general. We note that a positive ridge parameter can be incorporated into the bound, giving

$$\kappa \geq \frac{1}{2n} \left[ (\ell - n)\lambda_\ell + \delta + \sqrt{((\ell - n)\lambda_\ell + \delta)^2 + 4n\delta\lambda_\ell} \right]. \quad \square \tag{90}$$

We also note that, as observed by Jacot et al. (2020) and Spigler et al. (2020), the asymptotic scaling of $\kappa$ can be fixed if the kernel eigenvalues follow a power law spectrum. Specifically, if $\lambda_i \sim i^{-\alpha}$ for some $\alpha > 1$, then Jacot et al. (2020)[15] show that

$$\kappa = \Theta(\delta\, n^{-1} + n^{-\alpha}). \tag{91}$$

---

[15]Jacot et al. (2020) scale the ridge parameter proportional to $n$ in their definition of kernel ridge regression; our reproduction of their result accounts for this and applies to our convention.

