# OpenReview forum: "The Eigenlearning Framework: A Conservation Law Perspective on Kernel Ridge Regression and Wide Neural Networks"
_ICLR.cc/2023/Conference — Submitted to ICLR 2023_

### Official Review · Reviewer_F6bZ · 2022-10-25

**Confidence:** 2
**Correctness:** 2
**Technical Novelty And Significance:** 3
**Empirical Novelty And Significance:** 2
**Recommendation:** 6

**Clarity, Quality, Novelty And Reproducibility:**

The paper presents an original framework. Certain aspects looks so optimistic (like experiments of Fig.1), that I even start to doubt claims.

**Strength And Weaknesses:**

Among experimental results presented, the first one, described in Fig 1, seems to me the most valuable. The fact that even for moderate size widths, neural networks almost follow the same conservation of learnability, seems surprising (or this is true only for small n?). Why does this happen? IMHO, this should be discussed deeper, some additional comments are needed. Does this mean that for any architecture, from NTK one can correctly predict which functions are better learnable? This would be a very strong statement.

Section 6.1 is not self-contained. To understand it one needs to read citations given there. E.g. from the given explanation what is deep bootstrap phenomenon is not clear. The effective training time is not defined and etc.

In Section 6.2 it is claimed that the result generalizes  Bengio et al. (2006) to more general kernels. It would be interesting if authors discuss also, how different are required training set sizes according to Bengio et al. (2006) and in authors' analysis. How these two approaches are different.

Quantum mechanical analogy is not convincing.

**Summary Of The Paper:**

The paper presents a framework to describe the capacity of KRR to generalize different types of functions. An ability to generalize a function f is measured by the so called learnability (a measure that has a version that depends on training set inputs as well as a version that is expectation of the latter value). A key result is Theorem 3.2 that formulates a conservation of learnability. Authors claim that this result is similar in spirit with well-known “no-free-lunch” theorems of Wolpert.

**Summary Of The Review:**

Though some claims are not substantiated, a novelty of the framework is a strong side of the contribution.

---

> ### Author Response · Authors · 2022-11-09
> **Response to comments of Reviewer F6bZ**
>
> Thank you for your comments on our paper. We're pleased you agree with us that the experiments of Figure 1 are novel and interesting! It certainly surprised us that even moderately wide neural networks so closely obey the conservation law. One big question raised by our work is that of how informative these results are at finite width and for networks outside the NTK regime. We actually did take first steps towards exploring this empirically: we point the reviewer to Appendix C, in which we repeat the same experiment with increasingly narrow width. Indeed, as you say, the NTK correctly predicts which eigenfunctions are better learnable! We consider this a promising sign and an important message to bring to readers of ICLR. We certainly believe this experiment merits theoretical followup work.
>
> Our apologies that Section 6.1 is not self-contained. We were there limited by space. We will make efforts to clarify our exposition.
>
> Regarding our generalization of the result of Bengio et al. (2006), the result is of a similar nature - "exponentially many samples are required to learn the parity function with a certain class of kernel" - but the difference is that they restrict themselves to a particular kernel (the Gaussian kernel), whereas we allow a broader class of kernels (all rotation-invariant kernels, even those which are nonmonotonic).
>
> Would you mind elaborating upon your point that the quantum mechanical analogy is not convincing? Do you find that it seems invalid or merely that it is difficult to follow? We're fairly confident that it is correct, and the analogy runs surprisingly deep between these two canonical systems (though we can understand that this is regrettably difficult to appreciate without a physics background).
>
> Thank you again for your comments. If you feel we have adequately answered your questions and addressed your concerns, we’d humbly request you increase your score or reported confidence.

---

### Official Review · Reviewer_R9g9 · 2022-10-26

**Confidence:** 3
**Clarity, Quality, Novelty And Reproducibility:** No issue.
**Correctness:** 3
**Technical Novelty And Significance:** 3
**Empirical Novelty And Significance:** 3
**Recommendation:** 6

**Strength And Weaknesses:**

-) The paper is well-written and clear (apart from parts of the text which are made more complex than necessary using viewpoints borrowed from physics).
-) The experimental results are interesting and carry the main message with sufficient clarity.
-) Figure 1. Please explain what 1HL and 4HL mean.
-) The paper considers some concepts from a physics perspective which might be hard to follow for people without this background.
-) I do not fully understand why Theorem 3.2 is a "Conservation Law". My best understanding is the authors use this phrase because
the learnability remains fixed (in expectation) as the dataset evolves? This term is confusing, and rather un-necessary. For example, I can reach the global minimum of a strictly convex problem (assuming there is one) for any starting point. Is this a conservation law?
-) it would be nice to add more details regarding the training phase with eigenfuctions.

**Summary Of The Paper:**

The paper presents simplified  closed-form estimates for the test risk and other generalization
metrics of kernel ridge regression. Compared to prior work, the authors claim their derivations
are greatly simplified and lead to higher interpretability. Test risk and other objects of interest
are expressed in a transparent, interpretable way in terms of the conserved quantity evaluated
in the kernel eigenbasis. Among others, the main contributions include a theoretical explanation
for the “deep bootstrap” of Nakkiran et al. (2020), as well as a generalization of a result regarding
the hardness of the classic parity problem.

**Summary Of The Review:**

To the best of my knowledge the theory appears correct and the analysis reasonably novel. I choose "marginally above threshold" but I am happy to change my score if the authors provide more clarity.

---

> ### Author Response · Authors · 2022-11-09
> **Response to comments of Reviewer R9g9**
>
> We thank you for your comments on our paper. We are pleased that you found it largely novel, interesting, well-written, and clear. We largely agree with your constructive feedback – particularly your suggestions to improve the clarity of the manuscript – and have addressed it as follows.
>
> On 1HL and 4HL: these mean “single-hidden-layer” and “four-hidden-layer.” We have added this to the figure caption. Thank you for spotting the omission.
>
> On the use of physics terminology broadly: we appreciate you and other reviewers for giving us this feedback, and we have scaled back the reliance on physical concepts. In particular, we have rewritten Section 6.4 so as to shift jargon to the appendix and leave the main text readable to a general audience. Please let us know if anything in the main text remains unclear.
>
> Regarding the use of physics terminology: the terminology "conservation law" indeed stems from the fact that, as either the data distribution or the kernel changes, total learnability remains a constant. This is directly analogous to certain physical conservation laws, such as conservation of charge: no matter how the system changes, total charge remains constant. That said, we see how the term might be confusing, and we have added a sentence to Section 3 justfying it.
>
> Would you mind elaborating on your request for more detail regarding the "training phase with eigenfunctions"? Which section or experiment do you refer to? We are happy to do this.
>
> Thank you again for your feedback! We notice that you have stated that you’re happy to change your score if we address issues of clarity. If you feel we have adequately seen to your concerns, we’d humbly request that you do so!

---

### Official Review · Reviewer_zjAQ · 2022-10-28

**Confidence:** 4
**Correctness:** 3
**Technical Novelty And Significance:** 2
**Empirical Novelty And Significance:** Not applicable
**Recommendation:** 5

**Clarity, Quality, Novelty And Reproducibility:**

**Clarity & Quality:**
- The generalization error computation relies on a Gaussianity assumption (and a truncation heuristic) which is not mentioned in the main text. Appendix A only discusses the spectrum universality of kernel matrices, which does not necessarily imply the equivalence in the training/test loss. The authors should look into [Hu and Lu 2020] and references therein.
Hu and Lu 2020. Universality laws for high-dimensional learning with random features.

-  For MNIST and CIFAR data, can the authors comment on how the number of eigenvalues $M$ is selected, and how the eigencoefficients of the target function are computed from data?

**Novelty:** see weaknesses above.

**Reproducibility:** N/A.

**Strength And Weaknesses:**

## Strength

The derived generalization error formulae lead to a few interesting findings.

- A heuristic explanation of the deep bootstrap framework using the approximate equivalence between early stopping in gradient descent and ridge regularization.
- A learning lower bound for kernel methods on the parity problem.
- An analytic expression of the expected gradient norm of the estimator, which can be seen as a measure of robustness. To my knowledge this computation has not appeared in prior works.

## Weaknesses

My main concern is that the theoretical contribution in this submission is limited.

1. The generalization error formulae is derived from assuming the kernel features are Gaussian. This reduces the problem to linear regression on Gaussian features, for which the same set of equations has been rigorously proved and extensively studied in many prior works using random matrix theory, see [Hastie et al. 2019] [Wu and Xu 2020] [Richards et al. 2020]. These existing results are not thoroughly discussed in this submission, and I do not see the value of rederiving the same result using a non-rigorous approach (in fact, one could argue that the Stieltjes transform-based derivation is cleaner).

2. The "no-free-lunch" implied by the conservation of learnability is not surprising. Given a kernel with certain spectral bias, we should not expect it to perform well on all target functions. Moreover, analogues of "the recent folklore intuition that, given n samples, KRR and wide NNs can reliably learn at most O(n) orthogonal functions" have been rigorously shown in [Ghorbani et al. 2019] [Mei et al. 2021].

3. The applications of the error formulae in Section 6 are also not entirely new.
The explanation of "deep bootstrap" due to spectral filtering has already appeared in [Ghosh et al. 2021]. And the parity lower bound is a known result from various works such as [Daniely and Malach 2020] [Kamath et al. 2020].

4. The empirical validation of the generalization error formulae on neural network features has been done in [Loureiro et al. 2021]. Also, unlike the cross-validation estimators, the omniscient risk estimate studied in this submission requires access to "population" statistics, which can be computationally demanding in real-world applications.

Ghorbani et al. 2019.  Linearized two-layers neural networks in high dimension.
Hastie et al. 2019. Surprises in high dimensional ridgeless least squares interpolation.
Daniely and Malach 2020. Learning parities with neural networks.
Wu and Xu 2020. On the optimal weighted $\ell_2$ regularization in overparameterized linear regression.
Richards et al. 2020. Asymptotics of ridge (less) regression under general source condition.
Kamath et al. 2020. Approximate is good enough: probabilistic variants of dimensional and margin complexity.
Mei et al. 2021. Generalization error of random feature and kernel methods: hypercontractivity and kernel matrix concentration.
Loureiro et al. 2021. Learning curves of generic features maps for realistic datasets with a teacher-student model.
Ghosh et al. 2021. The three stages of learning dynamics in high-dimensional kernel methods.

**Summary Of The Paper:**

This submission computes the generalization error of kernel ridge regression (KRR) which depends on the eigenspectrum of the kernel and the decomposition of the target function onto the kernel eigenbases. The derivation is based on a Gaussian equivalence assumption, and the authors presented various interpretation of the formulae via the notion of learnability, which is the inner product between the KRR estimator and some given test functions. The theoretical predictions are empirically validated on synthetic and real-world datasets.

**Summary Of The Review:**

In my opinion this submission is below the acceptance bar due to the incremental theoretical contribution. I will consider updating my evaluation if the authors can adequately discuss the prior results and clarify the novelty in their theoretical analysis.

---

> ### Author Response · Authors · 2022-11-09
> **Response to comments of Reviewer zjAQ (part 1)**
>
> We thank you for your careful review of our work. We are pleased that you find the applications of our theory interesting, particularly our explanation of the deep bootstrap and estimator for robustness. We feel these applications are likely to have implications for the study of their respective problems and make a strong case for our paper’s interest to the ICLR community.
>
> We appreciate your constructive feedback. We feel we have appropriate responses to each of your main concerns via either edits to the manuscript or clarifying discussion and detail our responses below.
>
> **On the novelty of our estimators:** we thank you for the additional references. We were aware of [Hastie et al 2019] and cited it, and we have added citations to [Wu and Xu 2020] and [Richards et al. 2020] in the introduction and discuss them together with other works on linear regression with random features in Appendix A. We stress that our novel contribution re: these estimators is the new clarifying interpretation we give them, in terms of kernel eigenmodes competing for a fixed budget of learnability, and that this interpretation is new.
>
> A historical note is in order here: the generalization formulae we study have previously been studied in parallel by two camps, one studying KRR using tools from statistical physics and the other studying linear ridge regression using tools from random matrix theory. Despite obtaining similar results, these camps remain by and large unaware of each other. We intend for our paper to help bridge this gap by appropriately referencing both, and we thank you for your help in doing so.
>
> **Regarding the value of our derivation:** the value is of course not the fact that it is nonrigorous but rather that it is (a) more easily understood and (b) avoids the use of technical machinery, an achievement which is of interest in its own right.
>
> (a) Random matrix tools are complex and difficult to acquire, and most readers in the ICLR community will need to independently learn RMT tools before they can follow the Stieltjes-transform-based derivation. By contrast, ours should be parseable by virtually all those with a theory background. This *matters,* as people are hesitant to adopt results they do not understand. Given the premise that these generalization estimators are important results which should be widely appreciated, it is clearly of value to find a more accessible means to derive and express them, which is a central contribution of the present paper.
>
> (b) It is remarkable in its own right that we have obtained using simple linear algebra an important result which time and again required random matrix technology. We note that contributions of this sort, which rederive recent results by simpler means, are commonplace in mathematics and the physical sciences. Such contributions help to consolidate theoretical gains and illuminate the minimal path to results of interest. There is also clear inherent value to having a diversity of approaches. Suppose a later researcher wishes to extend these formulae, but finds that random matrix tools fail. Certainly it is then of value to know that these results can also be derived a different way which may prove more amenable to extension!
>
> **On the novelty of the no-free-lunch result:** we agree that [Ghorbani et al. 2019] and [Mei et al. 2021] show that, in some sense, “given n samples, KRR and wide NNs can reliably learn at most O(n) orthogonal functions.” This was a minor typo on our part: what we meant to say was that “given n samples, KRR and wide NNs can reliably learn at most **n** orthogonal functions.” These prior works show that O(n) functions can be learned, but leave the constant prefactor undetermined. The value (and the novelty) of bringing the conservation law into the discussion is that it is exact, nonasymptotic, and includes no constant factors, and thus it truly lets us make the above statement with n and not O(n).

---

> > ### Author Response · Authors · 2022-11-09
> > **Response to comments of Reviewer zjAQ (part 2)**
> >
> > **On the parity lower bound:** there have indeed been many results of the spirit that "the parity problem is hard for linear methods." *Our specific result, however, is new.* Our result states that the full (i.e. $d=k$) parity function is hard to learn for rotation-invariant kernels. Making this statement requires use of the eigenvalue ordering of the kernel to show that, as $k$ increases, the functions get monotonically harder to learn. By contrast, the prior results you mention rely on multiplicity arguments and thus cannot speak to the learnability of one particular function. Here are the differences in detail:
> >
> > * [Daniely and Malach 2020] use a multiplicity argument to show that parity functions of certain degrees are not learnable. However, their analysis is limited to $k \le \frac{d}{16}$ and relies on an adversarial choice of data distribution and finite-dimensional embedding space.
> > * [Kamath et al. 2020], if we understand it rightly, make a claim about the number of samples needed to learn **all** $2^d$ parity functions, not a specific one, which is unsurprisingly exponential.
> > * [Hsu 2020] (suggested by Reviewer zi9Y) use a multiplicity argument that does not apply at $k = d$.
> >
> > All considered, these analyses all lack the key ingredient for proving our result – the spectral bias of a rotation-invariant kernel – and thus their results are weaker. We have cited these works in the appropriate section.
> >
> > We note that the case $k=d$ is conceptually important because there is no degenerate subspace of similar functions to interfere with its learning, and yet it *still* proves hard to learn. We claim this paper is the first to show this special function is hard for generic rotation-invariant kernels.
> >
> > **On our explanation of the deep bootstrap:** While we appreciate the insight of Ghosh et al (2021), their hypersphere setting is unfortunately highly unrealistic, and their work is a far cry from explaining the deep bootstrap in general. One artifact of their artificial setting is that they find “three stages” in their training dynamics, while the deep bootstrap really only exhibits two. Realistic eigenvalues fall off as powerlaws, and the resulting training dynamics are qualitatively different. By contrast, our explanation manifestly applies to realistic data distributions, and we prove this definitively via experiments on real MNIST data.
> >
> > **On our spectral universality assumption:** we agree this should be mentioned in the main text and have made the appropriate edit.
> >
> > **On image eigendecompositions:** to be safe, we used $M \approx 10^4$, which is several times bigger than our maximal $n$ in these experiments. To compute eigencoefficients, we simply compute the $M \times M$ empirical kernel matrix, diagonalize it, and project the vector of targets onto the matrix eigenfunctions. Further details are in Appendix B.2.
> >
> > We thank the reviewer again for their feedback. We are committed to addressing the raised concerns – particularly missing discussion of prior work – and would be happy to field additional concerns. The review concluded with the suggestion that the final evaluation might be updated given an improved discussion of prior work and a clarification of the novelty and value of our approach. We believe we have done both. If the reviewer agrees, we respectfully ask that their score be updated accordingly.

---

### Official Review · Reviewer_zi9Y · 2022-10-28

**Confidence:** 4
**Correctness:** 4
**Technical Novelty And Significance:** 2
**Empirical Novelty And Significance:** 2
**Recommendation:** 6

**Clarity, Quality, Novelty And Reproducibility:**

Clarity:
I had a difficult time reading this paper, in part because it is unclear what is formally proved and what is not.

Quality:
The math I checked seemed correct.

Novelty:
The conservation law and the hardness of the parity problem are not novel. Furthermore, the adversarial robustness application could be explored some more, and the relation to Fermionic statistics needed some more motivation. On the other hand, the deep bootstrapping application seemed new.

Finally, there are already similar heuristics in the literature (Jacot et al. 2020, Canatar et al. 2021) for estimating the test MSE of kernel ridge regression. I am not sure why the one presented by this paper is preferable. It seems close to Canatar et al.'21.

**Strength And Weaknesses:**

## Strengths
This paper provides a convenient heuristic to predict the test MSE of a kernel ridge regression (and its bias and variance), as well as the train MSE. The authors use it to power several applications. The most interesting application is perhaps an explanation of the deep bootstrap effect . The authors' analysis extends results by Ghosh et al. 2022 that apply to spherically-distributed data, to any kernel with power law eigenvalue
decay.

## Weaknesses
There are some weaknesses that the authors should address.
### Level of formality
It is unclear what is proved and what is not. For example, in equations (6) through (14) it is not clear what is proved in the main text. The arguments in appendices H.6 and H.7 are informal, which is fine, but I feel that it should come with a disclaimer in the main text.

### Novelty of the framework
Be more specific in comparison to Canatar et al. '21 and Jacot et al. '20? It seems similar to the former.

### Novelty of the conservation law
The main conservation law, Theorem 3.2, essentially follows from the representer theorem for kernels:
given training data points $(x_i,y_i)\_{i \in [n]}$, kernel ridge regression will learn a function of the form $\hat{f}(x) = \sum_{i=1}^n \alpha_i K(x,x_i)$, for some scalars $\alpha_i$. Therefore, the best one can do to minimize MSE is to learn the projection of the target function $f$ onto this $n$-dimensional space $V$ spanned by the functions $K(x,x_i)$ for $i \in [n]$.

So (when $\delta = 0$) the "learnability" of $f$ is $\mathcal{L}(f) = \langle f, \Pi_V f\rangle = ||\Pi_V f||^2$. The conservation law states that for any
orthonormal functions $f_1,\ldots,f_m$ we have $\sum_{i=1}^m ||\Pi_V f_i||^2 \leq n$. However, this can be seen by writing $||\Pi_V f_i||^2 = \sum_{j=1}^n \langle v_j, f_i \rangle^2$, where $v_1,\ldots,v_n$ is an orthonormal basis of $V$, and switching order of summations. (Note this is essentially the same proof as remarked in Section G.1.)

The above proof is the basis for the kernel lower bounds technique surveyed in the note by Daniel Hsu https://www.cs.columbia.edu/~djhsu/papers/dimension-argument.pdf, which extends beyond $f_1,\ldots,f_m$ being orthonormal. It has been the basis for many recent lower bounds for kernels, and is not new.

### Application 2: Hardness of parity problem by rotationally-invariant kernels
This result is known (see the Daniel Hsu note). Note that it is true for permutation-invariant kernels (which are a superset of rotationally-invariant kernels).

### Application 3: Adversarial robustness
Please specify what the function $f$ is. Is it just $\mathrm{sgn}(x_1)$, or a smoothed version?

Predicted function smoothness does not seem to match experiment as you take $d$ larger in Figure 4. Why only test for $d \leq 8$, if the claim is about high-dimensional functions?

### Application 4: Investigation of relation between kernel ridge regression and quantum mechanics

This section (and Appendix F) has plenty of physics terminology and was difficult for me to understand. The connection to Fermionic particles seems to go through, but I don't understand what the takeaway is? The sigmoidal relationship between $\mathcal{L}_i$ and $\ln(\lambda_i)$ seems  apparent without having to appeal to physics. Simply because by definition $\mathcal{L}_i = \frac{\lambda_i}{\lambda_i + \kappa}$.



**Summary Of The Paper:**

This paper proposes a quantity called the "learnability" of a kernel, and shows that this satisfies a "conservation law". This is applied in four scenarios:

i) to explain "deep bootstrap" of Nakkiran et al
ii) to prove a result on the hardness of the parity problem by kernels
iii) to study adversarial robustness
iv) to investigate the relation between kernel ridge regression and quantum mechanics.

**Summary Of The Review:**

Unfortunately, because of the weaknesses I have written above, I do not recommend accepting the paper at this time.

---

> ### Author Response · Authors · 2022-11-09
> **Response to comments of Reviewer zi9Y (part 1)**
>
> Thank you for your comments on our manuscript. We are gratified by your positive evaluation of our explanation of the deep bootstrap and appreciate your extensive constructive criticism. We believe we can address a sizable fraction of it via a combination of new results, factual corrections, textual edits, and answers to your questions, and we hope this thereby changes your evaluation.
>
> **On formality:** thank you for raising this concern. We would like to be clear regarding the relative formality of our results. The answer is fairly simple: the results of Section 3 (on learnability) are formally proved, while the results of Section 4 (on the eigenlearning equations) are informally derived. We have added one sentence to each section stating this. (Edit: we've also clarified the discussion in Appendices H.6 and H.7 relative to the initial submission.)
>
> **On the novelty of the framework:** our main new contribution here is a *new interpretation* for the generalization behavior of KRR as an explicit competition between eigenmodes for the fixed budget of learnability. Downstream generalization metrics can all be expressed in terms of the learnability each mode gets (Eqns 6-14). This interpretation is simple, clarifying, powerful, and, as far as we can tell, new.
>
> To contrast directly with Canatar et al (2021): as far as what they allow one to compute, our results are basically the same as those of Canatar et al (2021) (except for certain edge cases with a discrete input space in which our framework applies and theirs does not; we discuss these in the appendix but choose not to emphasize them in the main text). We are happy to clarify this in the text if this was not clear. One should view our framework as a significant reformulation of prior results which makes them clearer and easier to derive, interpret and use. See the example comparison between our frameworks in the comment to all reviewers.
>
> While these estimators are numerically equivalent, ours admit interpretation more readily because we developed intuitive meanings for the learnabilities $\mathcal{L}_i$ and quantity $\mathcal{E}_0$.
> We encourage the reviewer to examine Appendix A, where similar comparisons are made using various estimators from other relevant works.
>
> This reformulation is a significant contribution because it makes these important results more accessible and much easier to grasp and use. (Our parity vignette demonstrates this: the same results can certainly be derived using the framework of Canatar et al (2021), but it requires some awkward algebra, while the result is trivial in our language.) Our derivations are also more accessible: most ICLR readers do not know the replica method and will not be able to follow the derivation of Canatar et al (2021), whereas ours should be parseable by virtually all those with a theory background.
>
> We note that our contribution here (i.e., simplification + clarifying reformulation of known results) is not of a typical type for an ML paper. Nonetheless, such work is important: it consolidates theoretical gains and makes them accessible to a broader audience. We found this particularly timely for recent work on KRR generalization, which has yielded powerful results but (we felt) would be more widely adopted and used if it were less technical to derive and more readily interpretable overall. This motivated the present study. We feel we have achieved these aims.
>
> The above suggests an important point to clarify for further constructive discussion: we posit that simplifying + shedding light on important known results is valuable for the community and that our work ought thus to receive favorable marks (despite the derivation of new estimators not being the primary goal of this part of the work).
>
> **On the novelty of our conservation law:** thank you for the reference to the note by Hsu. We have cited it accordingly and rephrased relevant claims. However, after exploring surrounding literature, we must stress that our application of this rule (to derive generally-applicable generalization estimators) is quite novel. If this rule is indeed already appreciated by a certain readership, *surely* it is of interest to this group that it can be leveraged to derive a full theory of generalization for KRR!

---

> > ### Author Response · Authors · 2022-11-09
> > **Response to comments of Reviewer zi9Y (part 2)**
> >
> >
> > **On the parity problem:** **Hsu’s result is not the same as ours!** Specifically, it is weaker. Set k = d in Hsu's note and you find that, since (d choose k) = 1, the sample complexity bound for learning the full parity function is vacuous. (This is because there indeed exists a permutation-invariant kernel targeted specifically towards each value of k, while this is not the case for rotation-invariant kernels.) In our case, it is nonvacuous (and exponentially large). One indeed needs to consider kernel eigenvalues to conclude that the full parity function is the hardest for a rotation-invariant kernel to learn, and this ingredient is missing from Hsu’s analysis. Given that this concern appears to be mistaken, we ask that it be removed from the count against us.
> >
> > (In comparing these two analyses, it is also worth noting that rotation-invariant kernels are much more common than permutation-invariant (but not rotation-invariant) kernels, which reflects favorably on our approach.)
> >
> > **On our study of function smoothness:** in our experiments, the function $f$ is a spherical harmonic of the specified $k$; this is merely a convenient choice, and one could also eigendecompose $\text{sgn}(x_1)$ and use that.
> > Thank you for your criticisms regarding our experiment. In response, we found and fixed a numerical bug behind the discrepancy and pushed the experiment to $d = 1024$. Agreement now improves at larger $d$, as expected from spectral universality results (and as necessary for the study of robustness). See the revised Figure 4. Having addressed your well-founded critiques, we ask this experiment be removed from the count against this paper and instead count in its favor.
> >
> > **On the KRR ≈ free Fermi gas mapping:** we apologize for the reliance on jargon here. This was an oversight, and we have rewritten the vignette so as to push physics terminology to the appendix and leave the main text broadly readable.
> >
> > As for "what is the point," there actually is a punchline we were working towards but were not able to finish before the deadline and have now added. Using this analogy, one can "solve" the implicit equation for $\kappa$, obtain an **explicit** expression for $\kappa$ of the form $\kappa = \text{[combinatorial sum over eigenvalues]}$ (see Equation 17 in the revision). We find this fairly remarkable, and, since studying the implicit equation generally requires indirect tools and bounds, we expect this explicit expression will prove useful in certain scenarios. Basically all related works in this area find implicit equations, but, as far as we know, we are the first to solve it for an explicit equation, which comes with obvious benefits. We ask that this count towards the novelty of our submission.
> >
> > As for other takeaways of the mapping, we find the universal curve collapse interesting in its own right, and since it is a special signature of KRR, it could be used to e.g. test whether an unknown kernel method is "KRR-like." (We acknowledge that we didn't need the physical analogy to see this, though, and have rephrased accordingly.) Furthermore, the mapping is itself of fundamental interest for ML researchers with a physics background (of which there are many in the ICLR readership): KRR and the free Fermi gas are two extremely well-studied canonical systems, and it is quite remarkable that their statistics turn out to be fundamentally the same.
> >
> > We thank you again for your useful feedback. We would be happy to field any additional questions or comments you have. Your review included a list of weaknesses you asked us to address; if you feel that we have addressed them to a reasonable degree, we would respectfully ask that you increase your score.

---

### Author Response · Authors · 2022-11-09
**Response to all reviewers (part 1)**

We thank the reviewers for their time and effort in reviewing our work. We are grateful for the positive and constructive feedback we received. We have accordingly made a number of edits to our manuscript, which are highlighted in blue. It is, however, our sense that the value of our work has regrettably not been fully appreciated by most reviewers, leading to our fairly low initial average score. This is understandable for several reasons, including that its significance is most apparent only with an intimate familiarity with certain aspects of related work, and we are optimistic about the prospects of constructive discussion with reviewers leading to a more favorable evaluation.

The dominant reviewer concern seems to be the novelty and significance of our work. There have been many related works studying the generalization of linear methods which arrive at similar risk estimates. What makes the present paper worthy of publication? This question has several compelling answers. Firstly, it gives a **novel interpretation of KRR** and an account of important results which is uniquely clear, insightful, and accessible compared to prior work, owing to the new manner in which they are derived, and secondly, **the four applications of our framework are each quite novel and interesting in their own right** to an extent which we believe was not fully appreciated by initial reviews. We expand on these points below.

The first point is short: we provide a novel interpretation of KRR as an explicit competition between kernel eigenmodes for learnability. Downstream metrics can be expressed entirely in terms of the amount of learnability each mode receives (Eqns 6-14). This picture is simple, clarifying, and, to the best of our knowledge, new.

The discussion around simplicity and accessibility is a bit longer, but it is important for appreciating the value of the work, so we include it.
Relative to prior work, the present work gives a simplified, more accessible treatment that offers additional insights. Our work was motivated largely by the observation that the many prior works on the subject seemed to use unnecessarily technical mathematical machinery (e.g. replica calculations and random matrix theory tools) in their derivations. While this may seem unimportant (particularly for one comfortable with such machinery), the use of complex tools tends to obscure intuition which is revealed by a simpler treatment. Accessibility is also important: we posit that, for most ICLR readers, understanding prior derivations will require first independently learning some replica or RMT tools (certainly this was the case for us), whereas ours can be understood with minimal background. This is important: people are hesitant to use results whose origins are hard to understand. If the prior works are important and interesting (as all reviewers seem to agree they are), then it is certainly of value to the ICLR community to clarify and simplify them. It is furthermore of *basic technical interest* that a calculation which has time and again required e.g. RMT tools can in fact be performed with ordinary linear algebra.

The remainder of our motivation was the observation that the final results of prior works could all be greatly simplified and summarized using a language which defines intermediate variables for how well each eigenmode is learned. Here is a typical illustration of this simplification. Here is the estimator for the covariance of the predicted function in the language of Canatar et al. (2021):

$$\sqrt{\eta_\alpha \eta_\beta} \text{Cov}\left[ w_\alpha^*, w_\beta^* \right] = \frac{1}{1-\gamma}\left(\sigma^2+\kappa^2 \sum_\rho \frac{\eta_\rho \bar{w_\rho}^2}{\left(P \eta_\rho+\kappa\right)^2}\right) \frac{P \eta_\alpha^2}{\left(P \eta_\alpha+\kappa\right)^2} \delta_{\alpha \beta}$$

and here is the same estimator in our language:

$$\text{Cov}\left[\mathbf{\hat{v}}_i,\mathbf{\hat{v}}_j\right] = \frac{\mathcal{E}(f) \mathcal{L}_i^2}{n} {\delta_i}_j.$$

Our version is not only more compact, it is far more interpretable because we have taken care (in Section 3) to develop intuition for the learnabilities $\mathcal{L}_i$. We think it is fairly uncontroversial that, while these estimators are numerically equivalent, the latter yields additional insight. (See Appendix A for other similar comparisons.) We generally claim that, while Canatar et al. (2021) and others deserve credit for first obtaining these estimators, in virtually all cases one should prefer our formulations. Our work is thus of interest to all authors who wish to apply these results to problems of interest.

---

> ### Author Response · Authors · 2022-11-09
> **Response to all reviewers (part 2)**
>
> More broadly, we claim that the fact that Equations (6-14) are *all* expressible solely in terms of modewise learnabilities, with no additional reference to eigenvalues required, demonstrates that modewise learnabilities are in some sense the “right” set of variables for an analysis of the generalization of KRR. We anticipate knowing this will help future studies into the subject extract useful intuition from their results and simplify their derivations, and the present study is thus of value to the ICLR community.
>
> We now discuss the four applications of our theory. We are gratified that all reviewers appreciated our explanation of the deep bootstrap phenomenon. We note that the *observation* of the deep bootstrap merited a paper in ICLR ‘21. It thus stands to reason that the *explanation* of this phenomenon merits publication in a major venue! This is especially true because, in deep learning as it stands today, interesting phenomena are common but convincing explanations are rare. We stress that our explanation’s short length in the main text may make it seem more easily dismissable, but, evaluating it as a contribution, we believe it to be significant. We believe we could have expanded this vignette into a full paper! We note that no reviewers questioned the correctness of our explanation.
>
> Regarding our treatment of the parity problem, we note that two reviewers mentioned similar prior works they claimed contained our result, but we believe that in fact they do not; see individual responses for details. We thank the reviewers for these references and have cited them.
>
> Our formulation of the mean squared gradient estimator breaks new ground and opens the way for new studies into both adversarial robustness. We note that the theoretical study of adversarial examples has been mostly stuck for some years, in large part because it is difficult to do any calculation which predicts robustness in a reasonable neural-net-like model. Our smoothness estimator breaks this impasse. If there is a chance that this idea could get the study of robustness unstuck, then *certainly* it is of interest to the ICLR community.
>
> (An additional point on the novelty of the above: despite having the tools to study more, all related prior works have restricted themselves to the sole study of error as the one quantity of importance. As far as we know, nobody has even bothered to experimentally *check* a KRR covariance prediction. We consider it a valuable meta-contribution to point out that it’s easy to imagine many other interesting things one can study with kernel eigenanalysis!)
>
> Finally, we believe our analogy to quantum mechanics is actually quite remarkable. KRR and the free Fermi gas are two highly canonical, very well-studied systems about which a great deal is known. (All physics undergraduates study the free Fermi gas.) It is thus surprising, and of clear fundamental interest, that their statistics are fundamentally the same (and that we just noticed after all this time)! We note that the fraction of the ICLR community with a background in physics is not small. This analogy is mostly written for that audience, but we accept reviewer feedback that that vignette ought to be written in a more accessible way and have restructured accordingly. As for fruit yielded by the analogy, in the updated manuscript we use our analogy to solve the implicit equation for $\kappa$ to obtain an *explicit* equation (see Equation 17 in the revision), which is of interest as all prior work is stuck with the implicit equation. This expression has not appeared in the KRR literature. This analogy is surprisingly deep – it is not forced – and it seems likely to bear other fruit.
>
> This is a fairly complete summary of the reasons we think our paper merits publication. Since most reviewer concerns dealt not with what was done but rather with why it is significant, we hope that, here and in individual responses, we have been able to convince reviewers of the value of the present paper. Please do reach out with further comments, and we look forward to continued productive exchange.

---

### Decision · Program_Chairs · 2023-01-20

**Decision:**

Reject

**Justification For Why Not Higher Score:**

The main results are not novel. While the presentation and formalism promise a simplification relative to prior approaches, the results themselves depend on unproven assumptions, and at least some reviewers did not find the claimed simplifications significant.

**Justification For Why Not Lower Score:**

N/A.

**Metareview: Summary, Strengths And Weaknesses:**

This paper studies generalization error and related metrics for kernel ridge regression and interprets the results through a number of quantities computed via the eigendecomposition of the kernel. The proposed framework enables discussion of the "deep bootstrap" and other phenomena, and a precise connection is drawn to the statistical physics model of a free Fermi gas.

During the discussion phase, reviewers highlighted the fact that the main theoretical results are not novel, to which the authors responded that their simplified interpretation and formalism were themselves of significant value, as were the four applications studied with their approach.

While it is sometimes true that a new presentation of old results can be of independent and substantial merit, in this case the added value is somewhat marginal. A main problem, as highlighted by reviewers, is that theoretical results here require a Gaussian equivalence assumption, which has been proven in certain related cases (such as some random feature settings), but remains unproven in the general setting considered here. A proof of this missing ingredient would likely require some of the advanced mathematical machinery that the authors have sought to avoid, thus undercutting the claimed benefits of their presentation, at least to some extent. A separate problem is that the claimed simplicity of presentation is somewhat subjective, and at least some of the reviewers did not seem to find benefit in this paper's presentation relative to the approaches in prior works.

Regarding the four applications, reviewers seemed to appreciate the discussion of the "deep bootstrap," but found the other applications less compelling. The authors make a strong case that establishing a connection to the statistical physics of a free Fermi gas is an important result, and I would agree. If there are novel insights made possible by this correspondence, I encourage the authors to highlight them in revision.